# Surface-atmosphere fluxes of volatile organic compounds in Beijing

W. Joe. F. Acton [1,a], Zhonghui Huang [2,b], Brian Davison [1], Will S. Drysdale [3,4], Pingqing Fu [5], Michael Hollaway [1,6], Ben Langford [7]; James Lee [3,4], Yanhui Liu [8,9], Stefan Metzger [10,11], Neil Mullinger [7], Eiko Nemitz [7], Claire E. Reeves [8], Freya A. Squires [3]; Adam R. Vaughan [3], Xinming Wang [2], Zhaoyi Wang [2], Oliver Wild [1]; Qiang Zhang [12], Yanli Zhang [2]; C. Nicholas Hewitt [1]

1.  Lancaster Environment Centre, Lancaster University, Lancaster, LA1 4YQ, UK
2.  State Key Laboratory of Organic Geochemistry and Guangdong Key Laboratory of Environmental Protection and Resources Utilization, Guangzhou Institute of Geochemistry, Chinese Academy of Sciences, Guangzhou 510640, China
3.  Wolfson Atmospheric Chemistry Laboratories, Department of Chemistry, University of York, York, YO10 5DD, UK
4.  National Centre for Atmospheric Science, University of York, York.
5.  Institute of Atmospheric Physics, Chinese Academy of Sciences, Huayanli 40, Chaoyang District, Beijing 100029, China
6.  UK Centre for Ecology & Hydrology, Library Ave, Bailrigg, Lancaster LA1 4AP, UK
7.  UK Centre for Ecology & Hydrology, Bush Estate, Penicuik, Midlothian, EH26 0QB, UK
8.  Centre for Ocean and Atmospheric Sciences, School of Environmental Sciences, University of East Anglia, Norwich, Norfolk, NR4 7TJ, UK
9.  Peking University, 5 Yiheyuan Road, Haidian District, Beijing 100871, China
10. National Ecological Observatory Network Program, Battelle, 1685 38th Street, Boulder, CO 80301, USA
11. Department of Atmospheric and Oceanic Sciences, University of Wisconsin - Madison, 1225 West Dayton Street, Madison, WI 53706, USA
12. Ministry of Education Key Laboratory for Earth System Modelling, Department of Earth System Science, Tsinghua University, Beijing, China

a. Now at: School of Geography Earth and Environmental Sciences, the University of Birmingham, Birmingham, UK

b. Now at: State Environmental Protection Key Laboratory of Environmental Pollution Health Risk Assessment, South China Institute of Environmental Science, Ministry of Ecology and Environment, Guangzhou 510655, China

*Correspondence to*: W. Joe F. Acton (W.J.F.Acton@bham.ac.uk)

**Abstract.** Mixing ratios of volatile organic compounds (VOCs) were recorded in two field campaigns in central Beijing as part of the Air Pollution and Human Health in a Chinese Megacity (APHH) project. These data were used to calculate, for the first time in Beijing, the surface-atmosphere fluxes of VOCs using eddy covariance, giving a top-down estimation of VOC emissions from a central area of the city. The results were then used to evaluate the accuracy of the Multi-resolution Emission Inventory for China (MEIC). The APHH winter and summer campaigns took place in November and December 2016 and May and June 2017 respectively. The largest VOC fluxes observed were of small, oxygenated compounds such as methanol, ethanol + formic acid and acetaldehyde, with average emission rates of $8.31 \pm 8.5$, $3.97 \pm 3.9$ and $1.83 \pm 2.0$ nmol m$^{-2}$ s$^{-1}$ respectively in the summer. A large flux of isoprene was observed in the summer with an average emission rate of $5.31 \pm 7.7$ nmol m$^{-2}$ s$^{-1}$. While oxygenated VOCs made up 60% of the molar VOC flux measured, when fluxes were scaled by ozone formation potential and peroxyacyl nitrate (PAN) formation potential the high reactivity of isoprene and monoterpenes meant that these species represented 30 and 28% of the flux contribution to ozone and PAN formation potential respectively. Comparison of measured fluxes with the emission inventory showed that the inventory failed to capture the magnitude of VOC emissions at the local scale.

## 1. Introduction

Air quality in urban areas is a pressing issue worldwide and is becoming the subject of much scientific, political and media focus. The Chinese capital Beijing is a megacity situated on the north China plain with a population of 22 million. Beijing suffers from periods of severe air pollution resulting from pollutant emissions occurring both within the city and from sources in the wider North China Plain region (Wehner et al., 2008). This pollution has a substantial impact on human health and economic output due to the large exposed population and the commercial and political importance of the city. For example, Gao et al. (2015) estimated that a single pollution event in January 2013, when the maximum hourly particulate matter of size smaller than 2.5 μm (PM$_{2.5}$) concentration in Beijing reached 650 μg m$^{-3}$, caused 690 premature deaths and ~250 million USD in economic losses.

Volatile organic compounds (VOCs) play an important role in local and regional air quality as they can oxidize and condense to form secondary organic aerosol (SOA) (Hallquist et al., 2009; Ehn et al., 2014) and because they take part in chemical reactions in the presence of NO$_x$ that can form ozone (Fehsenfeld et al., 1992). In addition, some VOCs, e.g. benzene, have direct detrimental impacts on humans and animals (e.g. Rinsky et al. (1987)). Urban areas, especially major cities, are important sources of these compounds into the atmosphere (Langford et al., 2009; Valach et al., 2015; Karl et al., 2018).

Several studies have reported VOC concentrations in Beijing, identifying alkanes, aromatics and oxygenated VOCs (Shao et al., 2009; Yuan et al., 2012; Wang et al., 2015). Comparison of air pollutant emissions between megacities (Parrish et al., 2009) showed that gasoline-fuelled vehicles were a major source of hydrocarbon

emission and in countries such as China industrial sources may also have a strong influence on hydrocarbon emission (von Schneidemesser et al., 2010). A positive matrix factorization (PMF) model applied by Wang et al. (2015) identified four major sources of VOCs in Beijing: two transportation factors denoted "gasoline evaporation and vehicular exhaust no. 1 (containing pentanes, acetylene, benzene, and toluene)" and "vehicular exhaust no. 2 (acetylene and C2-C4 alkanes)", "natural gas and liquid petroleum gas use and background" and "paint and solvent use and industry". Biogenic VOCs (BVOCs) represent an additional source of VOCs into the urban environment, the relative importance of which varies from city to city and from season to season. Ghirardo et al. (2016) used leaf-level emission measurements to estimate BVOC emissions in Beijing, including sesquiterpenes, benzenoids and fatty acid derivatives.

In most cities of the world it is challenging to directly observe the effectiveness of pollution control measures on ambient concentrations in real time. However, in Beijing the strict short-term emission controls applied during the 2008 Olympic Games (Liu et al., 2015) and the 2014 Asia-Pacific Economic Cooperation (APEC) summit (Li et al., 2017a) allowed the impact of pollution control measures in Beijing and the surrounding area on VOC concentrations to be investigated. Liu et al. (2015) reported that the emission controls during the 2008 Olympic Games caused a concentration reduction of 13 – 20% in a range of oxygenated VOC species but that acetone concentrations remained unaffected. Li et al. (2017a) found that when emission controls were applied to Beijing alone only a small decrease in VOC concentrations was observed but when controls also covered surrounding cities a large decrease (>40%) in most anthropogenic VOC concentrations was observed. This suggests that the atmospheric transport of pollutants from surrounding areas makes a large contribution to VOC concentrations in Beijing.

The impact of VOC emissions on local and regional air quality is examined using atmospheric chemistry and transport models, which rely on accurate estimates of the surface – atmosphere emission rates of pollutants. Much of the air quality modelling in Beijing relies on the Multi-resolution Emission Inventory for China (MEIC; available at http://www.meicmodel.org/, Qi et al. 2017) developed at Tsinghua University (e.g. Zhang et al., 2015; Hu et al., 2016). This relies on estimates of both source-specific emission factors and activity data and is hence a "bottom-up" estimation of emissions within a defined grid square. Validation of such an inventory is important in order to ensure they accurately represent real world emissions (Zhao et al. 2017).

Until comparatively recently it was very difficult to assess the accuracy of VOC emissions inventories at the district scale, but the development of proton transfer reaction-time of flight mass spectrometers now allows VOCs to be measured with sufficient time resolution for flux measurements to be made using the method of eddy covariance based on simultaneous micrometeorological measurements. Although the conditions under which eddy covariance can be applied in the heterogeneous city environment are somewhat limited by flow interferences from neighbouring buildings, night-time boundary layer stability and other considerations, the method has been successfully used for estimating both $CO_2$ fluxes (e.g. Song and Wang, 2012; Liu et al., 2012a; Song et al., 2013) and VOC fluxes (e.g. Velasco et al., 2005; 2009; Langford et al., 2009; Valach et al., 2015; Vaughan et al., 2017; Karl et al. 2018). However, VOC flux measurements have not previously been attempted in Beijing.

The work presented here was carried out as part of the "Sources and Emissions of Air Pollutants in Beijing" (AIRPOLL) project, itself part of the joint UK-Chinese "Air Pollution and Human Health in a Chinese Megacity"

(APHH) programme, described by Shi et al. (2019). The objectives of the study were to measure the surface-to-atmosphere fluxes of VOCs using the method of eddy covariance at a site in Beijing where a very comprehensive suite of atmospheric, physical and chemical parameters were assessed by the wider APHH-Beijing project team, and to use these data to evaluate the accuracy of a widely used emissions inventory. In this paper we therefore present the results of two intensive measurement campaigns covering both summer and winter conditions in Beijing. VOC mixing ratios were recorded using a proton transfer reaction time-of-flight mass spectrometer (PTR-ToF-MS) at a location in central Beijing. These data were then used, with three-dimensional wind velocity data, to calculate emission rates (fluxes) of VOCs, using the eddy covariance method (e.g. Müller et al., 2010) for the first time in Beijing. The calculated fluxes were then compared with the MEIC emissions inventory estimates for the measurement location.

## 2. Methodology

### 2.1. Site description

Measurements of VOC mixing ratios and fluxes were made during two intensive measurement campaigns (winter: 12/11/2016 - 10/12/2016; and summer: 15/05/2017 - 24/06/2017) from a meteorological mast located on the campus of the Institute of Atmospheric Physics, Chinese Academy of Sciences (IAP) in Beijing (39°58'33"N 116°22'41"E), 49 m above sea level. This site was used as the main sampling site for all APHH-Beijing programme activities (see Shi et al., 2019). The campus is situated north of central Beijing between the 3rd and 4th ring roads, with parkland to the east and west and a mix of dense residential and commercial (restaurants and shops) buildings to the north and south. The population density in the Chaoyang district is 7500 people km[-1] (Amir Siddique et al., 2020). Busy roads are situated 120 m north and 300 m east of the mast. Jing et al. (2016) classified the major roads present in the footprint as "Artery Roads". The average speed of roads in this classification peaked at ~43 km h[-1] at 04:00. Day time average speeds peak at ~33 km h[-1] (12:00) with a minimum average speed of ~28 km h-1 (18:00). Average traffic volume on artery roads peaked at ~3100 vehicles h[-1] (07:00) with a minimum average flow of ~250 vehicles h[-1] (03:00). In 2002 dominant tree species in Beijing were *Sophora japonica*, *Populus tomentosa* and *Juniperus chinensis* (Yang et al., 2005). Large scale tree planting occurred between 2012 and 2015 increasing forest cover by 10% with the main species planted being *Pinus tabulaeformis*, *Sophora japonica*, *Fraxinus chinensis*, *Salix spp.* and *Populus spp.* (Yao et al., 2019). For comparative purposes, the sampling location can be described as an "urban background" site with respect to ambient concentrations for central Beijing with the main sampling inlet being significantly elevated above ground level.

Flux measurements of $CO_2$ have been made previously at this site (Song and Wang, 2012; Liu et al., 2012a; Song et al., 2013). The fractional land cover within 2 km of the tower was assessed by Song and Wang (2012) and found to be buildings (0.65), vegetation (0.23) and roads (0.12). Using the local climate zone (LCZ) classification system described by Stewart and Oke, (2012) this area can be classified as LCZ2 "compact mid-rise" becoming LCZ1 "compact high-rise" to the south west. The aerodynamic roughness lengths were calculated by Li et al. (2003) as cited by Liu et al. (2012) and found to be 2.5, 3.0, 5.3 and 2.8 m from the north east, south east, south west and north west respectively with the zero-plane displacement heights of 12.3, 15.0, 26.4, and 13.2 m

respectively. The roughness length and displacement height at this site has remained consistent over the last ~20 years (Cheng et al., 2018b). The flux inlet was located at 102 m; this is over twice the mean height (45 m) of the tall residential building to the southwest (Liu et al., 2012a) and is below the mixing layer height (Squires et al., 2020) positioning the inlet in the constant flux layer.

## 2.2. Instrumental setup

A PTR-ToF-MS 2000 (Ionicon Analytik GmBH, Innsbruck) was housed in an air-conditioned container ~10 m from the base of the mast. The core operating principle of the instrument is described in detail by Jordan et al. (2009). The PTR-ToF-MS subsampled from two inlet systems, a common flux inlet line and a gradient switching system. The common flux inlet was 0.5 inch O.D. (I.D 3/8 inch) PFA and sampled air from ~50 cm below a sonic anemometer (Gill R3, Gill Instruments, Lymington, UK) which was mounted on a 3 m horizontal boom positioned on the mast at a height of 102 m above ground level. Air was drawn through the inlet line using a rotary vane pump (Model VT4.8; Becker, Hull, UK). The flow rate in the inlet line was recorded using a mass flow meter (TSI Mass Flowmeter 4043, TSI, Shoreview, USA) and remained in the range 90 - 103 L min$^{-1}$ during both campaigns. At 90 L min$^{-1}$ the Reynolds number in the inlet line is 12900, indicating turbulent flow. Particulates were removed from the air flow by a 90 mm Teflon filter installed near the inlet. This filter was replaced each day.

To minimise the impact of the tower on flux measurements, the sonic anemometer was positioned so that air from the prevailing wind directions would not be disturbed by the tower. The sonic anemometer was installed on the north west side of the tower in the winter and the south east in the summer. Analysis of the turbulence characteristics (Squires et al., 2020) did not show an impact from the structure even when the air flow passed through the tower. This is likely due to the open lattice construction of the tower and the predominance of larger eddy scales at 102 m.

Gradient measurements were made by switching between five sampling heights (3, 15, 32, 64 and 102 m) and zero air. Air was drawn down five separate 0.25 inch O.D. (I.D. 1/8 inch) PFA lines using a common second rotary vane pump (Model VT4.8; Becker, Hull, UK). For each height, the flow was sampled at 3.3 L min$^{-1}$ of which ~300 ml min$^{-1}$ was drawn through a 10 L stainless steel container and an additional bypass flow of 3 L min$^{-1}$ was used to reduce the residence time in each line (Fig. 1). The five containers were heated to 30 °C to limit adsorption/desorption effects and had a turn-over time of approximately 30 minutes. For two minutes each hour the PTR-ToF-MS sequentially subsampled from the flow exiting each container via a manifold valve, providing hourly 30-minute average mixing ratio at each of the five heights. The PTR-ToF-MS was operated in an hourly cycle switching between the flux and gradient measurements. The PTR-ToF-MS sampled from the gradient system for the first twenty minutes of each hour and from the flux line for the final 40 minutes of the hour. A zero air generator was built in house and made up of a platinum catalyst heated to 260 °C, and zero air was sampled for 5 minutes each hour at the end of the gradient cycle.

The PTR-ToF-MS was operated with an inlet flow rate of 30 sccm and an $E/N$ ratio (where $E$ is the electric field strength; $N$ is the buffer gas density) of 130 Td. To achieve this, the drift tube was maintained at 60 °C with a pressure 1.9 mbar and a voltage of 490 V applied across it. Data were acquired by the PTR-ToF-MS at a 5 Hz

time resolution, allowing VOC fluxes to be calculated using the eddy covariance method. In order to facilitate

mass calibration, trichlorobenzene was introduced by diffusion through a needle valve into the inlet stream.

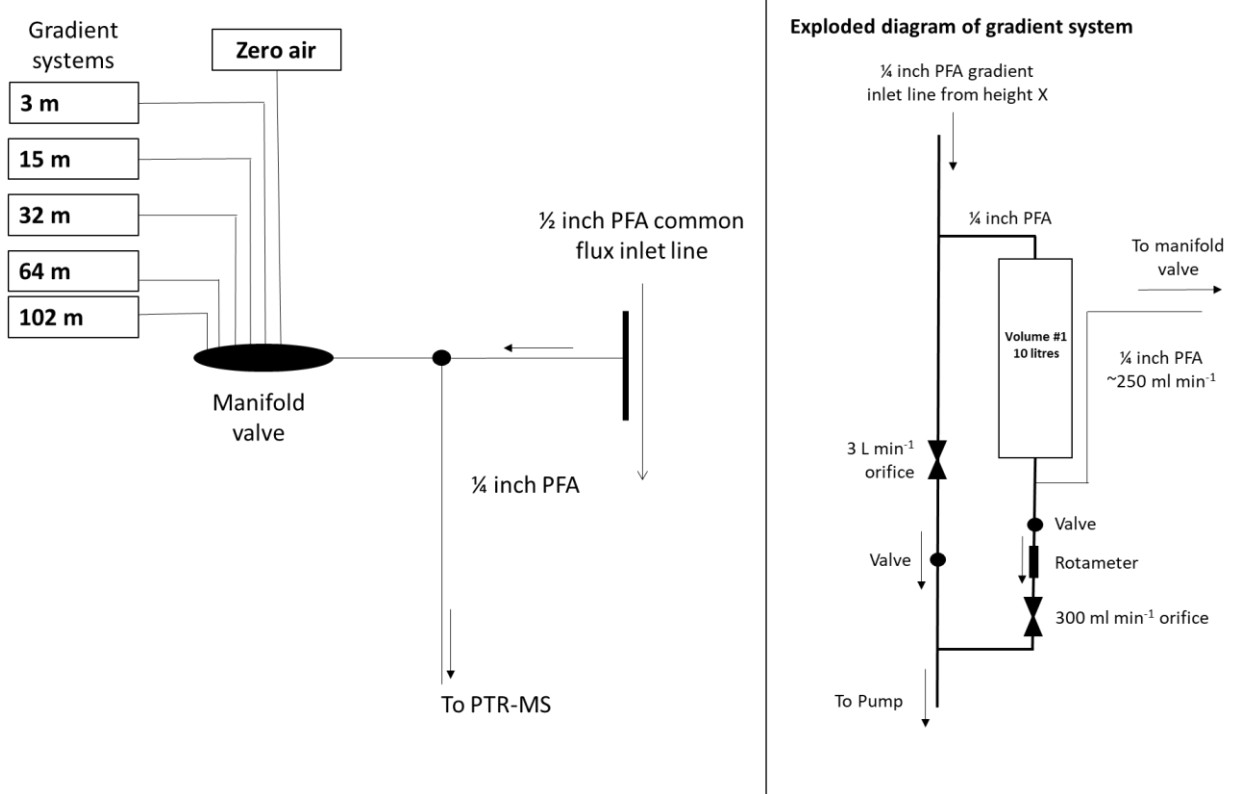

Figure 1: Schematic of the gradient sampling system. The PTR-ToF-MS inlet was switched between a common flux inlet line (40 min) and a gradient switching manifold (20 min) in an hourly cycle. The gradient system sampled from five heights into 10 stainless steel containers.


### 2.3. Calibration

The PTR-ToF-MS was calibrated twice weekly using a 15 component VOC standard (National Physical Laboratory, Teddington, UK) during both the winter and summer campaigns. This standard contained methanol, acetonitrile, ethanol, 1,3-butadiene, acetone, isoprene, butenone, butan-2-one, benzene, toluene, m-xylene and

1,2,4-trimethylbenzene at $1 \pm 0.1$ ppmv each. The standard was dynamically diluted in zero air to provide a six point calibration. The sensitivities to these compounds in the winter and summer campaigns are summarised in the supplementary information.

During the winter campaign, the instrument was also calibrated three times using a second independent 15 component VOC standard (Ionicon Analytik, Innsbruck, Austria) containing methanol, acetonitrile, acetaldehyde,

ethanol, acrolein, acetone, isoprene, crotonaldehyde, butan-2-one, benzene, toluene, o-xylene, chlorobenzene, α-pinene and 1,2-dichlorobenzene at $1 \pm 0.05$ ppmv each. At the end of the campaign the PTR-ToF-MS was also calibrated using an aldehyde standard (Ionicon Analytik, Innsbruck, Austria) made up of formaldehyde, acetaldehyde, acrolein, propanal, crotonaldehyde, butanal, pentanal, hexanal, heptanal and octanal at $1 \pm 0.05$ ppmv each, nonanal at $600 \pm 0.07$ ppbv and decanal at $500 \pm 0.08$ ppbv. Calibration using both Ionicon standards

was performed following dynamic dilution in zero air to give a six point calibration. In the summer campaign, four point calibrations were performed using both Ionicon VOC standards once a week.

Over the course of the winter campaign 29 h (6% of the PTR-ToF-MS operational period) were lost to instrument maintenance and calibration and 133 h (16% of the PTR-ToF-MS operational period) were lost to instrument maintenance and calibration in the summer.

**2.4. Volume mixing ratio calculation**

PTR-ToF-MS mass calibration and peak fitting was performed on the 5 Hz data using PTRMS Viewer 3 (Ionicon Analytik, Innsbruck). VOC mixing ratios were then calculated using the method based on that previously applied by Acton et al. (2016), Tani et al. (2004) and Taipale et al. (2010). Counts per second (cps) of each protonated VOC species ($RH^+$) were normalized against the primary ions ($H_3O^+$ and the $H_2O.H_3O^+$ cluster ion) and

background counts were subtracted to give background corrected normalised count rates $I(RH^+)_{norm}$:

$$I(RH^+)_{norm} = I(RH^+)\left(\frac{I_{norm}}{I(H_3O^+)+I(H_3O^+H_2O)}\right) \quad\quad (1)$$

$$-\frac{1}{n}\sum_{i=1}^{n} I(RH^+)_{zero,i}\left(\frac{I_{norm}}{I(H_3O^+)_{zero,i}+I(H_3O^+H_2O)_{zero,i}}\right),$$

where $I(H_3O^+)$ and $I(H_3O^+H_2O)$ represent the measured count rate for $H_3O^+$ and the $H_3O^+H_2O$ cluster, respectively. Zero air measurements were labelled using the subscript zero and the number of zero air measurements was represented by $n$. The total reagent ion count rate ($H_3O^+$ and $H_3O^+H_2O$) was normalised to a count rate of $10^6$ cps ($I_{norm}$).

Following calculation of the background corrected normalised count rate, the volume mixing ratio (*VMR*) was

calculated, as a wet mass fraction, as:

$$VMR_{VOC} = \frac{I(RH^+)_{norm}}{S_{norm}}, \qu\quad\quad (2)$$

where $S_{norm}$ is the normalised sensitivity (ncps/ppbv) for each mass calculated using a transmission curve as described by Taipale et al. (2008). Mixing ratios of specific compounds were then determined by summing parent ion and fragment ion mixing ratios. For direct comparison with the molar flux, the volume mixing ratio was

converted to a molar concentration ($\chi_{VOC}$) using equation 3.

$$\chi_{VOC} = \frac{P \times VMR_{VOC}}{R \times T}, \qu\quad\quad (3)$$

where $P$ is the atmospheric pressure, $T$ is atmospheric temperature and $R$ is the molar gas constant (8.314 J K$^{-1}$ mol$^{-1}$). Note that in the following presentation of results and discussion, the abundance of VOCs in air are described in terms of volume mixing ratio (parts per billion; ppbv) although strictly speaking they should be

described as the wet mass fraction (with the same units).

## 2.5. Flux calculation

The continuous fast measurement of PTR-ToF-MS for all compounds allows the estimation of fluxes using direct eddy covariance rather than the virtual disjunct eddy covariance methods that had to be applied when using older PTR-quad-MS instruments (e.g. Karl et al., 2002; Langford et al., 2009; Velasco et al., 2005; 2009). Eddy covariance fluxes of VOCs using PTR-ToF-MS were first made by Müller et al. (2010). A detailed description of flux measurement using eddy covariance and its limitations is provided by Aubinet et al. (2012). Here fluxes were calculated using the eddy4R routines (Metzger et al., 2017; R Core Team, 2019) with further analysis performed using the open air and ggmap packages (Carslaw and Ropkins 2012; Kahle and Wickham, 2013). The flux ($F_x$) of each compound was determined by calculating the covariance function between the vertical wind velocity ($w$) and the VOC concentration ($\chi_x$; nmol m$^{-3}$):

$$F_x(\Delta t) = \frac{1}{N}\sum_{i=1}^{N} w'(i - \Delta t/\Delta t_w)\chi_x'(i), \tag{4}$$

where $w'$ and $\chi'$ represent momentary deviations from the mean concentartion or vertical wind speed (i.e. $w' = w - \overline{w}$). $N$ is the number of PTR-ToF-MS measurements in our half hour averaging window (9000 for 5 Hz measurements), $\Delta t_w$ is the sampling interval between wind measurements (0.2 s) and $\Delta t$ is the lag time between the vertical wind measurements recorded by the sonic anemometer at 102 m and the concentrations recorded using the PTR-ToF-MS instrument at ground level.

The lag time between the vertical wind velocity measurement and VOC concentration measurement is primarily controlled by the inlet line length and the flow rate in the line. However, small variations in temperature, pressure, humidity and pump performance, and also horizontal displacement between anemometer and inlet coupled with changes in wind speed and direction can all cause deviation in the lag time. The lag time can be determined by assessing the covariance between $w$ and $\chi$ as a function of time. The lag time can then be identified by selecting the maximum of this covariance function (Taipale et al., 2010). For many of compounds recorded in this study only a weak flux was observed. This results in a low signal-to-noise ratio, introducing a large uncertainty into the identification of the cross-covariance maximum. Adsorption and desorption rates to the inlet line are compound specific and dependent on polarity. Therefore, the lag times for more polar compounds such as oxygenated VOCs will differ slightly from pure hydrocarbons but weak covariance peaks made the identification of the lag-times difficult for many masses. Lag times were therefore determined by calculating the lag for compounds where a strong covariance peak was observed (isoprene and C2-benzene ($C_8H_{10}$) in the summer and winter campaigns respectively) and applying their modal lag times as a fixed value to all masses. The calculation of lag time for eddy covariance data with low signal-to-noise ratio is described in detail by Langford et al. (2015).

The effect of storage below the measurement height was calculated for flux averaging period $t$ as:

$$Storage\ F_{x\,t} = h\frac{\chi_{x\,t-1} - \chi_{x\,t+1}}{3600}\ . \tag{5}$$

Where $\chi_{x\,t-1}$ and $\chi_{x\,t+1}$ are the concentration (nmol m$^{-3}$) of compound $X$ in the averaging periods before and after flux averaging period, $t$, and $h$ represents the measurement height in meters (102 m). This method was chosen so as to be comparable with the other flux measurements made during this project (e.g. Squires et al. 2020). This storage term was then added to the calculated turbulent flux term.

At the 102 m measurement height, large eddies mean that some of the flux may not be captured in the 30 min flux averaging period used here. This loss of low frequency flux was investigated by Squires et al. (2020) who found that at the 102 m measurement height the flux loss was ca. 7%. Due to similarity of the transport mechanisms underlying the low frequency flux loss (e.g., Mauder et al., 2020) we assume this estimate to apply also to the simultaneously measured VOC fluxes presented here. We did not correct the fluxes presented in this paper for low frequency flux loss. We further investigated the effect of high-frequency spectral loss on the VOC fluxes using a wavelet-based methodology (Nordbo and Katul, 2013). Loss of high frequency flux due to measuring at 5 Hz was estimated to be less than 10%, which we did not correct for in the presented fluxes. Fluctuations in air density caused by changes in temperature and humidity can impact fluxes (Webb et al., 1980). As a closed path system was used in this study, the impact of air density variations caused by changes in sensible heat flux are negligible but variations in air density caused by changes in the latent heat flux may impact the VOC flux calculation. The impact of changes in latent heat flux on the $NO_x$ flux at this site was assessed by Squires et al. (2020) and found to be significantly less than 1% throughout both campaigns. As the sensitivity of a trace gas flux to density fluctuations is controlled by the ratio of its constituent density over its flux (Pattey et al., 1992) this ratio was assessed for the measured VOC species. This ratio was found to be comparable to that observed for $NO_x$ by Squires et al. (2020), suggesting that the uncertainty in VOC flux as a result of changes in latent heat flux was less than 1%. Therefore the Webb-Pearman-Leuning correction (WPL: Webb et al., 1980) for changes in the density of air to the calculation of fluxes was not applied. Squires et al. (2020) estimated that the average time taken for an air parcel to reach the inlet point at 102 m was ~68 s. The OH concentration in the summer campaign reached $2.8 \times 10^7$ molecules cm$^{-3}$ (Whalley et al., 2020) and the ozone concentration $2.6 \times 10^{12}$ molecules cm$^{-3}$, at these concentrations ~10% of monoterpenes (the most reactive species recorded) would have reacted before the measurement height.

Half hour averages were quality-assessed using three tests after a 2-dimensional coordinate rotation to correct for tilting of the sonic anemometer. The limit of detection was calculated for each mass by determining the cross-covariance between 150-180 s, a region outside the expected time lag range (Spirig et al., 2005). In order to calculate a reliable flux, the averaging period must meet stationarity requirements such that this period is shorter than the time scale at which changes in atmospheric conditions occur. The stationarity of the flux across the half hour averaging period was assessed using the method described by Foken and Wichura (1996). The half hour averaged fluxes were filtered where the stationarity criterion (the difference between average flux and mean average flux of its components) exceeded 60% and were above the limit of detection as for fluxes below the limit of detection a robust statement on their stationarity could not be made. Lastly, files were also flagged if the mean frictional velocity ($u_*$) was less than 0.175 m s$^{-1}$, this threshold was derived from assessment of the $u_*$ dependence of the sensible heat flux. Data falling below this threshold were substituted by the campaign average value for that hour so as not to introduce a positive bias to the VOC flux. The random error of each half hour flux average was assessed using the method described by Lenschow et al. (1994) and fluxes with a relative random error greater than 150%, averaged across the campaign, were discarded.

In the winter campaign, 25% of the flux averaging periods fell below the $u_*$ filter. Of the masses contributing more than 0.75% to the total flux, the average percentage of flux files filtered for failing the stationarity test is 69%. This ranged from 73% for methyl vinyl ketone and methacrolein formaldehyde to 62% for acetaldehyde. In

the summer field campaign 12% of the flux averaging periods fell below the $u_*$ filter. Of the masses contributing more than 0.75% to the total flux, the average percentage of flux files filtered for failing the stationarity test is 25%. This ranged from 52% for formaldehyde to 13% for methanol.

## 3. Results and Discussion

### 3.1 Meteorology

During the winter measurement campaign (7[th] November to 10[th] December), wind speeds were low, ranging between 0.3 and 9.7 m s$^{-1}$, with a mean value of 2.4 m s$^{-1}$, and with the highest wind speeds observed from the North West (Fig. 2). The predominant wind directions were southerly and north westerly. Temperatures, measured at 102 m on the IAP tower, ranged from -7 to 15 °C with a mean value of 3.6 °C. Relative humidity ranged from 15 to 92% with a mean value of 45% (Fig. 3). Precipitation was small during the campaign period with light rain on 20[th] November and snow on 21[st] November.

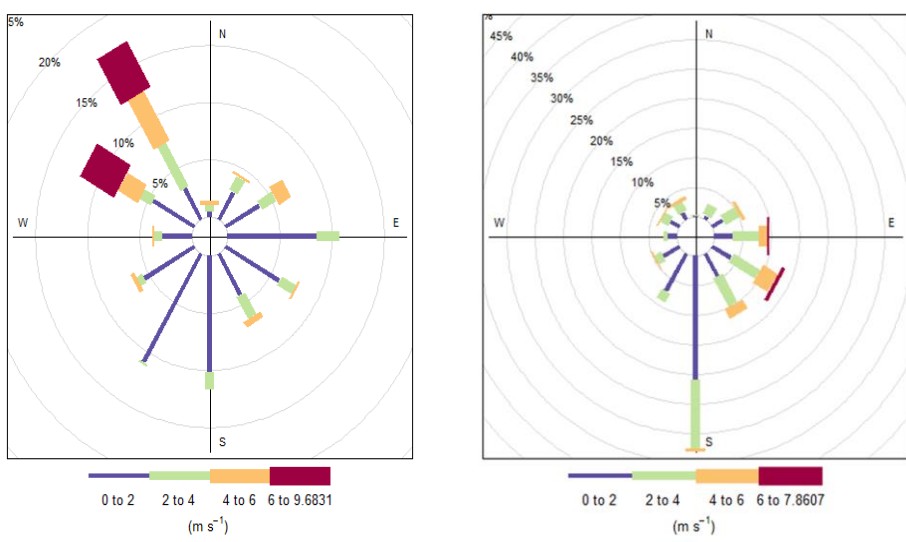

Figure 2: Wind conditions during the winter campaign (left) and summer campaign (right). The plots show the percentage of time the wind was from each direction and are coloured by wind speed.

During the summer measurement campaign (20[th] May - 22[nd] June 2017) wind speed ranged between 0.5 and 7.9 m s$^{-1}$, with a mean value of 2.3 m s$^{-1}$. The predominant wind direction was south-easterly (Fig. 2). Temperatures at 102 m ranged from 15 to 37 °C with a mean value of 25 °C and the relative humidity ranged from 13 to 93% with a mean value of 44% (Fig. 3). Precipitation was low during the campaign period with light rain observed on 22[nd] and 30[th] May and on 2[nd], 6[th], 18[th] and 21[st] – 23[rd] June.

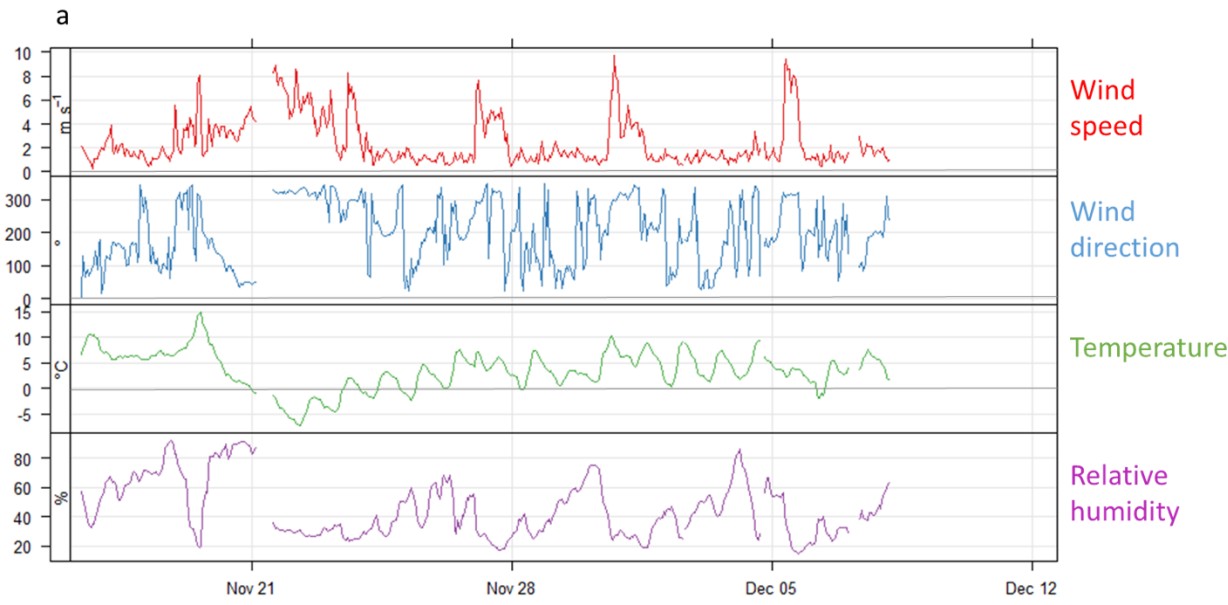

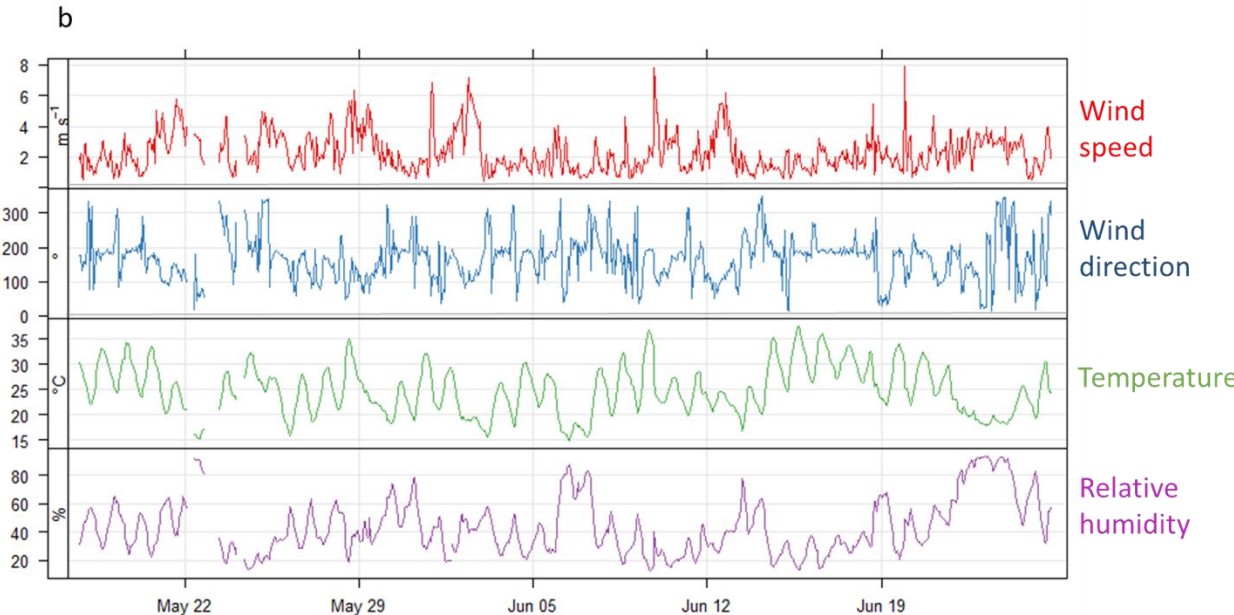

Figure 3: Summary of meteorological data in winter measurement campaign (a) and summer measurement campaign (b).

### 3.2 Flux footprint

The flux footprint was calculated for each half hour flux averaging period of both field campaigns using the method described by Kljun et al. (2004) and Metzger et al. (2012). The calculation of the flux footprint for this campaign is described in detail by Squires et al. (2020). During both the summer and winter field campaigns 90% of the measured flux originated from an area within 7 km of the IAP tower; however, 90% of the contribution to the campaign average flux footprint extended just 2 km from the tower. Mean flux footprint climatologies for the

summer and winter campaigns are displayed in Fig. 4, with contour lines showing the distances from the tower

where the surface contributions to the measured fluxes cumulate to 30, 60 and 90%, respectively. In the winter campaign the main contribution to the average flux was predominantly from the northwest and southeast, encompassing two large roads: the Jingzang Expressway and the Beitucheng West Road and a mix of commercial and residential buildings, and urban park land. In the summer campaign, the largest contribution to the flux came from regions approximately 1 km south west of the tower and approximately 1 km north east of the tower now

encompassing different sections of the Jingzang Expressway, Beitucheng West Road, residential buildings, commercial buildings (shops and restaurants) and a larger contribution from urban park land compared to the winter campaign.

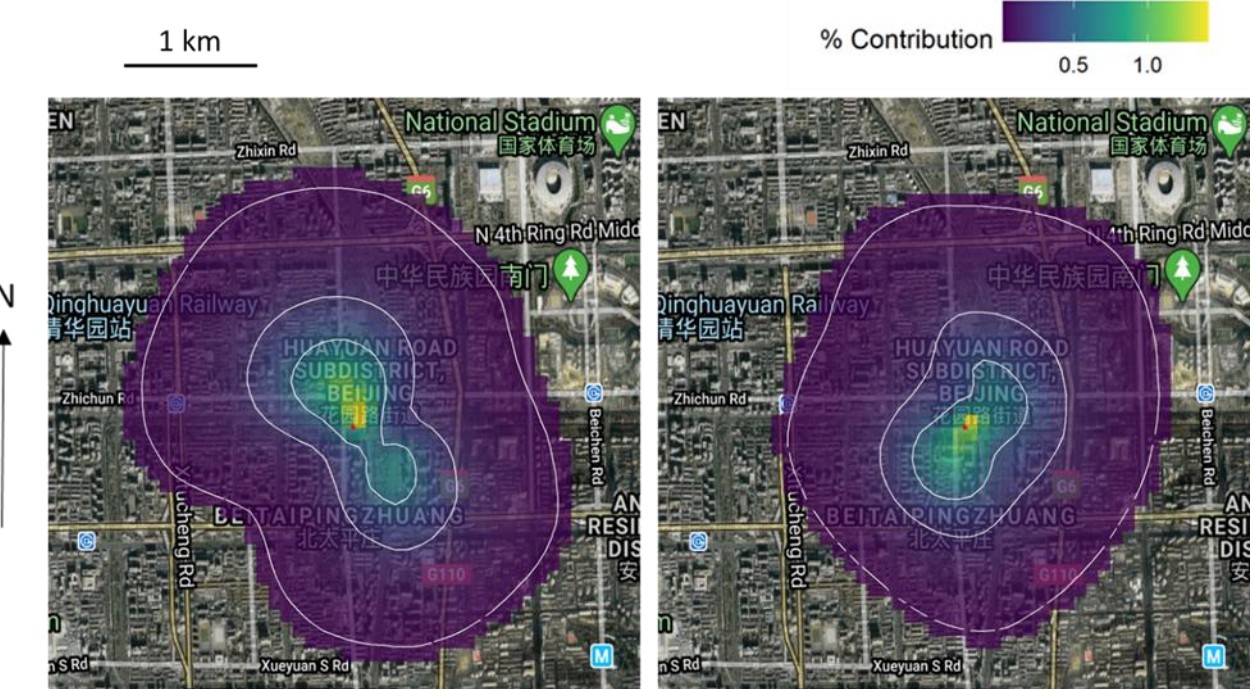

Figure 4. The campaign mean flux footprint climatologies for the winter (left) and summer (right) field campaigns. The IAP meteorological tower is represented by the red dot and surrounding $100 \times 100$ m cells are coloured by their mean contribution to the flux. The contour lines correspond to the distances from the tower where the surface contributions to the measured fluxes cumulate to 30, 60 and 90% respectively. Map tile sets are © Google. Map data @ 2020 Imagery @ 2020, CNES/Airbus, Landsat/Copernicus, Maxar Technologies.

### 3.3. VOC mixing ratios

In the winter campaign the detected VOC species with the largest mixing ratios were the small oxygenated VOCs (OVOCs) methanol, acetaldehyde and acetone, with median mixing ratios of 19.29, 4.59 and 2.57 ppbv, respectively. Large mixing ratios of aromatic compounds were also observed, with median mixing ratios between

1.5 and 2.0 ppbv for each of benzene, toluene and $C_2$-benzenes. Mixing ratios of these compounds tracked the pollution events in a distinctive "sawtooth cycle" described by Jia et al. (2008) where mixing ratios increased over

a period of 3 - 4 days when the wind speed was low and prevailing wind direction was southerly before dropping rapidly when the wind direction moved to the North West. In the summer campaign, the dominant VOC species observed were again methanol, acetaldehyde and acetone (26.39, 3.50 and 3.67 ppbv respectively), but the mixing ratios of aromatic compounds were lower, with median values of 0.30, 0.37 and 0.47 ppbv observed for benzene, toluene and $C_2$-benzenes respectively. Methanol and ethanol + formic acid together account for approximately one third of the total oxygenated VOC mixing ratio. These compounds have been observed at high concentrations in many cities (Langford et al., 2009 (Manchester; summer); Shao et al., 2009 (Beijing; summer); Valach et al. 2015 (London; August-September); Sahu and Saxena 2015 (Ahmedabad; winter)) with ethanol reported to be the most abundant VOC in London with a mean mixing ratio of 5 ppbv in summer and winter (Dunmore et al., 2016).

Isoprene, which has both biogenic and vehicular sources (Borbon et al., 2001; Jaimes-Palomera et al., 2016), had median mixing ratios of 1.08 and 0.38 ppbv in the winter and summer campaigns respectively. High winter isoprene concentrations have been observed previously in Beijing (Li et al., 2019) where an average value of 1.01 ppbv was observed as well as during haze periods (Sheng et al., 2018). High winter isoprene has also been reported in other Asian countries with isoprene mixing ratios of 1.6 and 1.1 ppbv recorded in Ahmedabad (Sahu and Saxena 2015) and Kathmandu (Sarkar et al., 2016). These studies are all characterised by the use of a PTR-MS, it is possible that an isomer of isoprene contributes to the high winter time concentrations observed in Asian cities. In contrast, other studies have reported higher summer time isoprene concentrations in Beijing (e.g. Cheng et al., 2018a). In the winter, isoprene mixing ratios peaked during haze events and correlated well with anthropogenic and oxygenated VOCs. $R^2$ values of 0.82, 0.77 and 0.74 were observed for the correlation of isoprene with propenal, benzene and toluene respectively.

In contrast to the winter campaign, mixing ratios of most compounds during the summer campaign showed a clear diurnal cycle with mixing ratios peaking at night and dropping during the day as the planetary boundary layer expanded during the day-time and contracted at night-time (Fig. 5). A summary of VOC mixing ratios for the principal VOC species observed during the two campaigns are displayed in Table 1. For a more detailed discussion of VOC mixing ratios recorded during this campaign see Zhang et al. (2020).

Table 1. Summary of VOC mixing ratio for the dominant VOCs observed in Beijing during APHH-Beijing winter and summer campaigns (in ppbv).

| | Methanol | Acetonitrile | Acetaldehyde | Acetone | Isoprene | Benzene | Toluene | $C_2$-Benzenes |
|---|---|---|---|---|---|---|---|---|
| *Winter campaign* | | | | | | | | |
| *Max* | 93.98 | 2.13 | 17.42 | 9.66 | 4.50 | 9.11 | 9.43 | 13.07 |
| *Min* | 0.00 | 0.00 | 0.22 | 0.00 | 0.00 | 0.00 | 0.00 | 0.00 |
| *Median* | 19.29 | 0.38 | 4.59 | 2.57 | 1.08 | 1.59 | 1.65 | 1.65 |
| *Mean* | 24.32 | 0.51 | 4.92 | 2.80 | 1.21 | 2.00 | 1.94 | 2.08 |
| *Standard deviation* | 20.83 | 0.51 | 3.66 | 2.00 | 1.03 | 1.74 | 1.87 | 2.07 |
| *Summer campaign* | | | | | | | | |
| *Max* | 73.75 | 24.44 | 26.92 | 11.76 | 3.24 | 1.89 | 2.74 | 3.41 |
| *Min* | 7.10 | 0.00 | 0.00 | 0.70 | 0.00 | 0.00 | 0.00 | 0.00 |
| *Median* | 26.39 | 0.29 | 3.50 | 3.67 | 0.38 | 0.30 | 0.37 | 0.47 |
| *Mean* | 28.39 | 0.97 | 4.64 | 3.86 | 0.56 | 0.37 | 0.46 | 0.60 |
| *Standard deviation* | 11.02 | 1.86 | 3.57 | 0.44 | 0.55 | 0.31 | 0.37 | 0.52 |

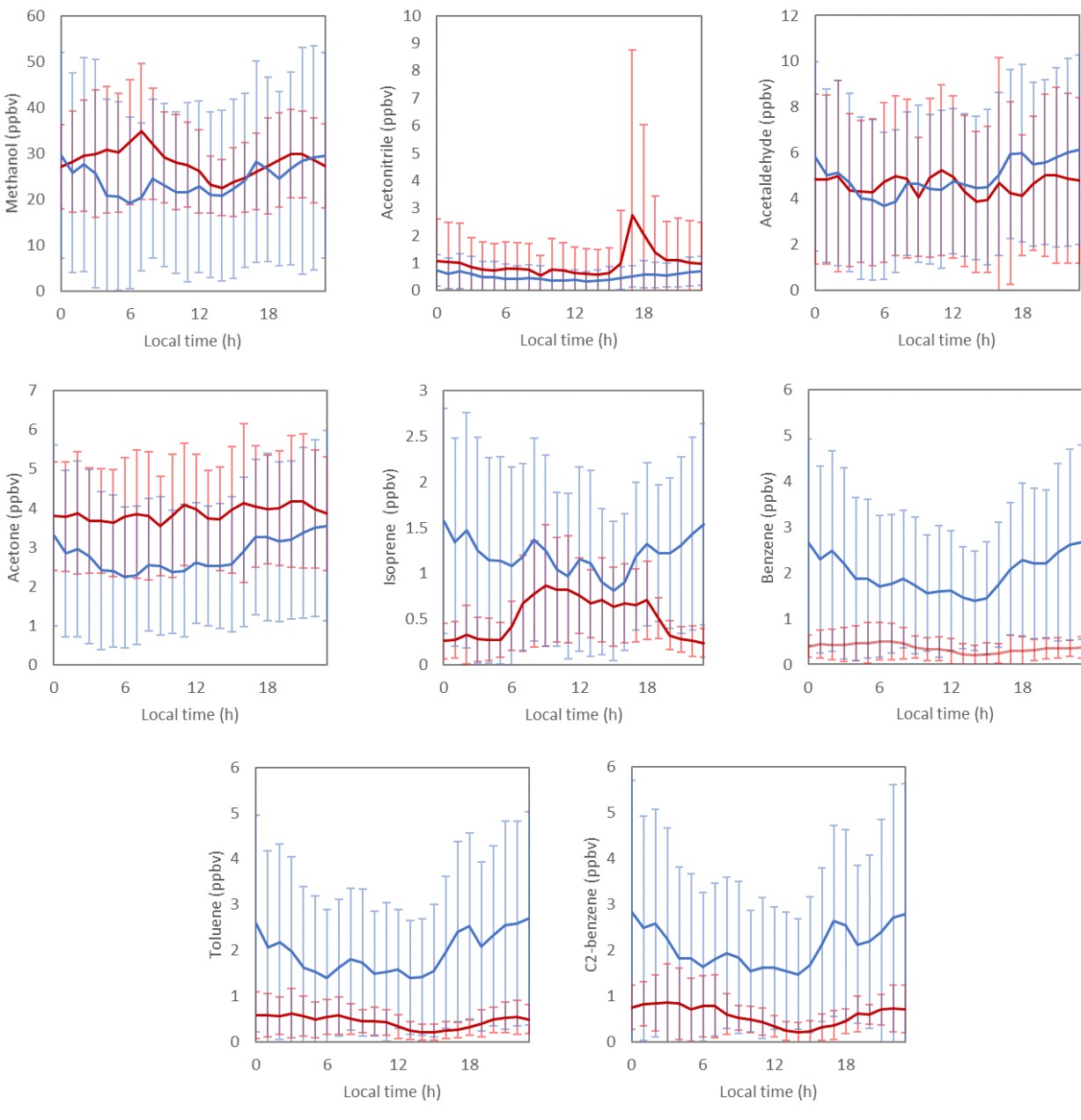

Figure 5. Diurnal profiles of the dominant VOCs observed in Beijing during APHH-Beijing winter (blue) and summer (red) campaigns (in ppbv). Error bars show standard deviation across the measurement period.

### 3.4 VOC fluxes

The mixing ratio values represent the ambient concentrations observed at the sample inlet averaged over the sampling period. However, by combining these mixing ratios with the three dimensional wind vector, the eddy

covariance flux calculation method allowed for quantification of the net exchange (flux) of the observed compounds from a surface "footprint" to the point of sampling. The size of this footprint depends on the height of measurement and surface roughness as well as a number of meteorological factors including wind speed and atmospheric stability. Under the conditions in Beijing, 90% of the campaign average measured flux originated from an area within 1-2 km of the tower (see Fig. 4). The fluxes observed at 102 m are therefore predominantly

controlled by local emissions, unlike the mixing ratios observed at the same point which are influenced by both

local and more distant emissions, chemical processing and meteorology. Fluxes of $CO_2$ have been previously recorded at the IAP field site (Liu et al., 2012a; Song and Wang, 2012; Song et al., 2013). Liu et al. (2012a) reported four years of $CO_2$ eddy covariance measurements showing a seasonal variation in emissions driven by winter heating and uptake of $CO_2$ by vegetation in the summer. Spatial variation in emission was shown to be determined by the surface cover and the diurnal profile of emissions was largely dependent on traffic volumes.

Greater surface solar heating in the summer led to stronger turbulence than was observed in the winter, with median $u_*$ values of 0.41 and 0.26 m s$^{-1}$ respectively. This meant turbulent transport was more easily measured, allowing the fluxes of a larger number of compounds to be quantified in summer than in the winter. Of the 179 masses observed in the summer campaign, 64 masses gave a flux with a median relative random error of less than 150%. These masses all showed a net positive flux with no masses showing net deposition. Molecular formulae were assigned to 50 of these masses using their exact mass, while 15 masses with a small negative mass defect could not be allocated a formula, based on the combination of C, H, N, O, S and Cl within 0.1 mDa of the recorded mass. Mass spectral peaks observed at m/z 47.019 and 47.053, assigned to formic acid and ethanol, respectively, could not be sufficiently resolved so these compounds were considered together.

Diurnal cycles of the six compounds with the largest fluxes are shown in Fig. 6 (solid lines), showing the large VOC flux observed in the summer campaign, in comparison to the much smaller fluxes observed in the winter campaign. This figure also displays fluxes without the storage term applied (dashed lines) and the gap-filled $u_*$ filtered flux (dotted lines). As shown, these corrections have only a small impact on the fluxes calculated. The fluxes of these six compounds are summarised in Table 2. As shown in Table 2, fluxes of all compounds were larger in the summer than in the winter. This was not the case for $NO_x$ and CO which were larger in the winter than in the summer (Squires et al., 2020). There may be several reasons why the measured VOC fluxes were lower in winter than summer. In the winter, the high concentrations of VOCs at this site, mainly resulting from advection from outside the city, might lead to deposition of VOCs. This would have the effect of suppressing the net VOC emission rate measured at the site. In the winter campaign OH concentrations peaked at $8 \times 10^6$ cm$^{-3}$ (Slatter et al., 2020) and the average $O_3$ concentration was 16.4 µg m$^{-3}$ (Shi et al., 2019). Using rate coefficients from Atkinson and Arey (2003) the expected atmospheric lifetimes of benzene and 1-butene with respect to OH would be 1.2 days and 1.1 h, respectively. The lifetimes of benzene and monoterpenes with respect to $O_3$ would be over a year and 6 days, respectively. The lifetime of less reactive compounds such as aromatics is therefore long enough to allow for transportation from outside the flux footprint. The average temperature in the winter campaign was 3.6 °C compared to 25 °C in the summer, and this would result in lower volatilisation of liquid VOCs (for example in gasoline), supressing the emission rates. Sources such as cooking and solvents in cleaning products, cosmetics and paints contribute to urban VOC emission (Karl et al., 2018) and these sources are likely to be reduced in the winter when cold weather causes ventilation of buildings to be reduced. Measured VOC emissions from indoor and outdoor sources may therefore be lower in winter. In addition the lower atmospheric turbulence in winter and generally lower fluxes resulting in greater uncertainty in our winter fluxes compared to summer fluxes. Our detailed analysis, therefore, focuses on the fluxes recorded during the summer campaign.

Table 2: Summary of (storage corrected) fluxes of VOCs displayed in Table 1 including the six dominant VOC fluxes recorded in Beijing during APHH-Beijing winter and summer campaigns (in nmol m$^{-2}$ s$^{-1}$).

| | Methanol | Acetonitrile | Acetaldehyde | Formic acid + Ethanol | Butene | Acetone | Acetic acid | Isoprene | Benzene | Toluene | C$_2$-Benzene |
|---|---|---|---|---|---|---|---|---|---|---|---|
| | | | | | Winter campaign | | | | | | |
| Max | 56.37 | 1.22 | 11.45 | 8.45 | 2.58 | 3.87 | 11.02 | 26.01 | 4.49 | 3.71 | 4.60 |
| Min | -48.63 | -0.62 | -7.21 | -7.05 | -1.77 | -2.24 | -6.77 | -8.17 | -2.93 | -2.64 | -2.54 |
| Median | 1.12 | 0.05 | 0.04 | 0.57 | 0.10 | 0.03 | 0.07 | 0.21 | 0.07 | 0.16 | 0.00 |
| Mean | 1.87 | 0.10 | 0.53 | 0.86 | 0.13 | 0.15 | 0.29 | 0.67 | 0.23 | 0.22 | 0.17 |
| Standard deviation | 13.33 | 0.27 | 2.24 | 2.24 | 0.64 | 0.78 | 2.56 | 3.33 | 0.96 | 0.77 | 0.93 |
| | | | | | Summer campaign | | | | | | |
| Max | 55.36 | 10.64 | 11.89 | 53.67 | 16.42 | 7.65 | 14.25 | 59.52 | 2.79 | 18.16 | 12.76 |
| Min | -21.33 | -17.63 | -3.57 | -4.19 | -1.23 | -1.81 | -3.07 | -2.04 | -1.05 | -0.76 | -0.99 |
| Median | 6.40 | 0.16 | 1.43 | 3.00 | 1.00 | 0.75 | 1.28 | 2.20 | 0.25 | 0.51 | 0.51 |
| Mean | 8.31 | 0.22 | 1.83 | 3.97 | 1.57 | 0.97 | 1.67 | 5.31 | 0.33 | 0.86 | 0.84 |
| Standard deviation | 8.49 | 1.56 | 1.96 | 4.90 | 2.01 | 1.06 | 2.15 | 7.69 | 0.33 | 1.50 | 1.33 |

**3.4.1 Total measured VOC flux**

In the summer campaign, the total measured VOC flux was strongly positive with the city centre acting as a net source of VOCs into the atmosphere. In the winter campaign there was a small net positive flux of most identified VOC species but at times deposition of VOCs was also observed (Fig. 6). The total measured VOC fluxes and concentrations during the summer field campaign, coloured by compound class, are displayed in Figs. 6a and 6b respectively, with the total measured VOC flux peaking at $80 \pm 24$ nmol m$^{-2}$ s$^{-1}$ at mid-day. As alkanes have a

proton affinity less than that of water these compounds cannot be measured using PTR-MS so are not included here.

Methanol and ethanol + formic acid together also make up one third of the total molar flux. The mean methanol flux recorded in the summer campaign was comparable to that observed in two spring time (April and March) campaigns in Mexico City where 9.1 and 12.8 nmol m$^{-2}$ s$^{-1}$ of methanol were observed respectively (Velasco et

al. 2005; 2009). The winter methanol flux was closer to the 2.5 nmol m$^{-2}$ s$^{-1}$ reported by Valach et al. (2015) in London (August-December). Other oxygenated compounds (grouped as oxygenated VOCs in Fig.7) together make up another 31% of the summer flux. This is consistent with the large flux of oxygenated VOCs observed by Karl et al. (2018) from the urban canopy in Innsbruck, Austria. Fluxes of acetone were also recoded in London (1.5 nmol m$^{-2}$ s$^{-1}$; Valach et al. 2015) and Mexico City (1.9 nmol m$^{-2}$ s$^{-1}$; Velasco et al. 2005) both of which were

higher than the $0.15 \pm 0.78$ and $0.97 \pm 1.06$ nmol m$^{-2}$ s$^{-1}$ measured in the summer and winter campaigns in this study. Unclassified compounds, labelled "other chemical groups" in Fig. 7 were made up primarily of masses corresponding to compounds containing nitrogen, sulphur and halogens and make up 15% of the total concentration but contribute only 9% to the total flux. The largest fluxes in this group were an unidentified mass at m/z 40.96 and m/z 87.03 ($[C_4H_6S]H^+$). The (non-oxygenated) hydrocarbons have been divided into three

categories: predominantly-biogenic compounds (isoprene and monoterpenes), aromatic compounds and alkenes. Together these compounds make up 15% of the total VOC mixing ratio observed but make up 31% of the total measured VOC flux, with predominantly-biogenic compounds the largest contributors.

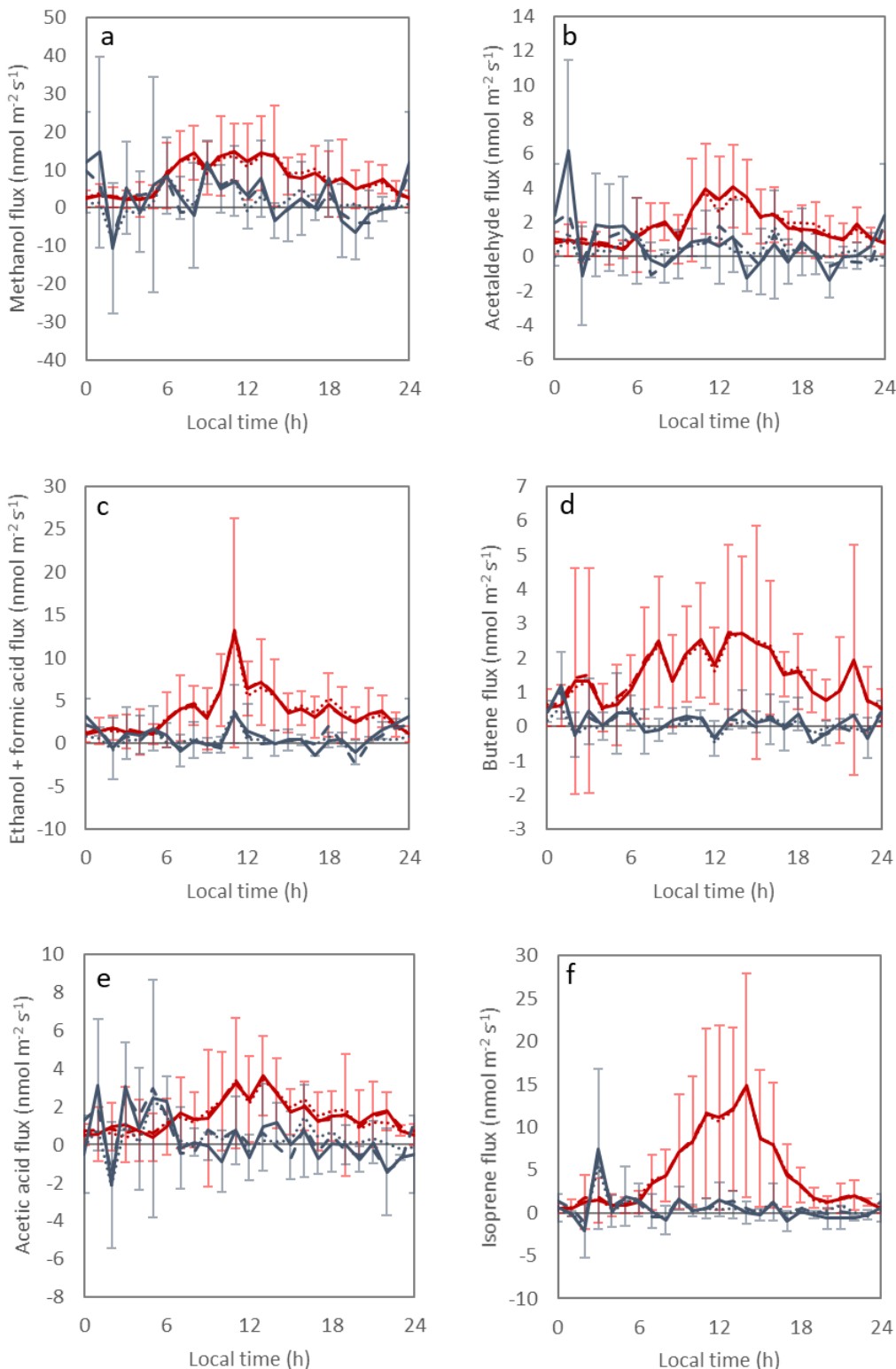

Figure 6: Diurnal cycles of the fluxes of six compounds with the largest observed fluxes for the summer (red) and winter (blue) campaigns, with error bars showing standard deviation of the stationarity filtered and fluxes including the storage term (solid lines). Dashed lines represent the flux without the storage term and dotted lines are $u_*$ filtered fluxes including the storage term and gap filled with the average flux above the $u_*$ threshold for that hour.

**3.4.2 Anthropogenic VOC fluxes**

Combustion products are a major source of anthropogenic VOCs in urban areas. The correlation between (a) the $NO_x$ flux and (b) those VOC species contributing more than 0.75% to the total measured VOC flux is shown in Fig. 8. $NO_x$ fluxes during the campaigns have previously been reported by Squires et al. (2020). In urban areas NOx is almost exclusively produced by combustion sources of which motor vehicles are the main source. The NOx flux is therefore used here as proxy for vehicular emissions. The fluxes of the aromatic VOCs toluene, $C_2$-benzenes (the sum of xylenes and ethyl benzene which cannot be separated using PTR-MS) and $C_3$-benzenes (the sum of all compounds featuring a benzene ring and three methyl groups) show a good correlation with the fluxes of $NO_x$, with $R^2$ values of 0.75, 0.64 and 0.61, respectively. Fluxes of toluene, $C_2$ and $C_3$ benzenes all follow a similar diurnal profile: after a small peak in the early morning emissions increase from 06:00 and peak between 10:00 and 15:00 and then decrease in the evening with another small peak at 22:00. As shown in Fig. 8 the fluxes of aromatic compounds were closely correlated, suggesting that they are emitted from the same source. Beijing operates a motor vehicle emissions control program which only allows non-local heavy duty vehicles to enter the city between 00:00 and 6:00 (Yang et al., 2015) and this may explain the peak in aromatic emission observed at 02:00 (Fig. 7). These compounds are known to be combustion products and are also emitted by evaporation. These fluxes are discussed in detail by Squires et al. (2020) who found that while the winter benzene and toluene fluxes ($0.23 \pm 0.96$ and $0.22 \pm 0.77$ nmol m$^{-2}$ s$^{-1}$, respectively) were smaller than those observed in London and Manchester, the Beijing summer fluxes ($0.33 \pm 0.33$ and $0.86 \pm 1.50$ nmol m$^{-2}$ s$^{-1}$ for benzene and toluene, respectively) were comparable to those reported in Europe (Langford et al., 2009; Langford et al., 2010; Valach et al., 2015; Vaughan et al., 2017). The toluene flux is, however, significantly lower than the 2.5 and 9.2 nmol m$^{-2}$ s$^{-1}$ recorded at two sites in Mexico City (Velasco et al. 2005; 2009). Velasco et al. (2009) identified both exhaust emissions and evaporative sources contributing to the toluene flux. The addition of these evaporative sources in Mexico City as well as differences in vehicle fleets may explain why these fluxes were larger than those recorded in this study and in the UK. Squires et al. (2020) also concluded that the benzene/toluene ratio was within the range expected for primary exhaust emissions with values for concentrations of 0.89 and 0.73 in the winter and summer respectively. These values are, however, higher than the 0.6 calculated by Barletta et al. (2005) for Beijing. For fluxes, the benzene/toluene ratio was 0.72 in winter and 0.31 in summer (Squires et al. 2020). The summer value is lower than expected for primary exhaust emissions, suggesting that fuel evaporative loss contributes to aromatic VOC emission in summer (Squires et al. 2020).

The fluxes of propene and methanol were shown to correlate with the flux of $NO_x$ with $R^2$ values of 0.63 and 0.71. Propene is known to be emitted by vehicular traffic (Velasco et al. 2005; 2009) and the correlation with $NO_x$ suggests that traffic is the main source at this site. Propene has also been recorded in plumes of VOCs released from industrial activity (Karl et al., 2003), but no large industrial sites were present within the flux footprint of the IAP tower. Large fluxes of propene were recorded, with average emissions of $0.75 \pm 3.2$ and $0.93 \pm 1.1$ nmol m$^{-2}$ s$^{-1}$ in the winter and summer, respectively. In the winter the strongest emission occurred early in the morning with little emission later in the day. In the summer measurement period, emissions increased from 06:00 before peaking at mid-day and decreasing in the late afternoon. Methanol is emitted from plants and industry and is formed by oxidation reactions in the atmosphere (Jacob et al., 2005). It is also present in many consumer goods products (Steinemann, 2015). The close correlation with the $NO_x$ flux suggests a combustion source contributes to the total methanol flux but the summer methanol flux also correlates well with small oxygenated VOCs such

as acetaldehyde (Fig. 8). Methanol had the largest molar flux of any VOC species recorded in the summer (Fig. 6), with average emissions of $1.87 \pm 13.3$ and $8.31 \pm 8.5$ nmol m$^{-2}$ s$^{-1}$ in the winter and summer, respectively. These methanol emissions are comparable to those observed in London and Mexico City with 2.5 nmol m$^{-2}$ s$^{-1}$ (August to December) recorded by Valach et al. (2015), 8.2 nmol m$^{-2}$ s$^{-1}$ (October) reported by Langford et al. (2010) in London, and 9.1 and 12.8 nmol m$^{-2}$ s$^{-1}$ (April and March respectively) reported in Mexico City (Velasco et al., 2005; 2009). Methanol mixing ratios were, however, significantly higher in Beijing than in London with average mixing ratios in Beijing 28.2 and 24.3 ppbv in the winter and summer compared with 7.5 and 19.4 ppbv reported by Valach et al. (2015) and Langford et al. (2010) at 61 and 200 m, respectively.

Acetonitrile is considered a tracer for biomass burning (de Gouw et al., 2003), but can also be produced by the burning of fossil fuels (Holzinger et al., 2001). Acetonitrile fluxes in the summer field campaign were low, with a median value of $0.16 \pm 1.6$ nmol m$^{-2}$ s$^{-1}$, but regularly peaked between 16:00 and 18:00 in the evening with a maximum value of 10.6 nmol m$^{-2}$ s$^{-1}$, potentially indicating a cooking source. In the winter the median acetonitrile flux was small ($0.05 \pm 0.3$ nmol m$^{-2}$ s$^{-1}$) with peaks of up to 1.22 nmol m$^{-2}$ s$^{-1}$ observed in the early hours of the morning. As cooking rates are likely to be comparable in the summer and winter, the reduced winter acetonitrile flux could be explained by reduced ventilation of homes in the winter.

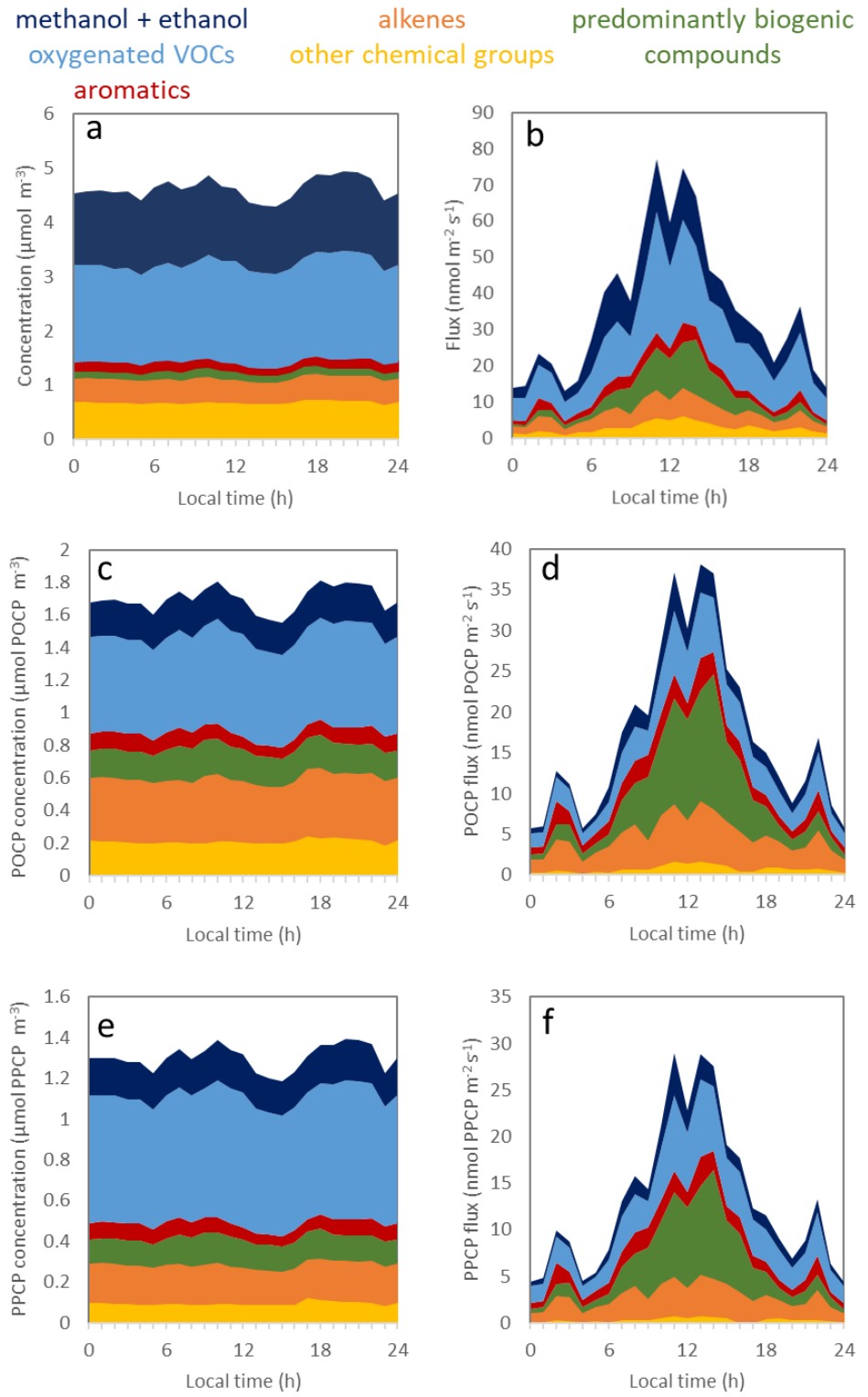

Figure 7. Total observed VOC concentration (a) and flux (b) measured during the summer field campaign, scaled using photochemical ozone creation potentials (POCPs) (c, d) and photochemical PAN creation potentials (PPCPs) (e, f).

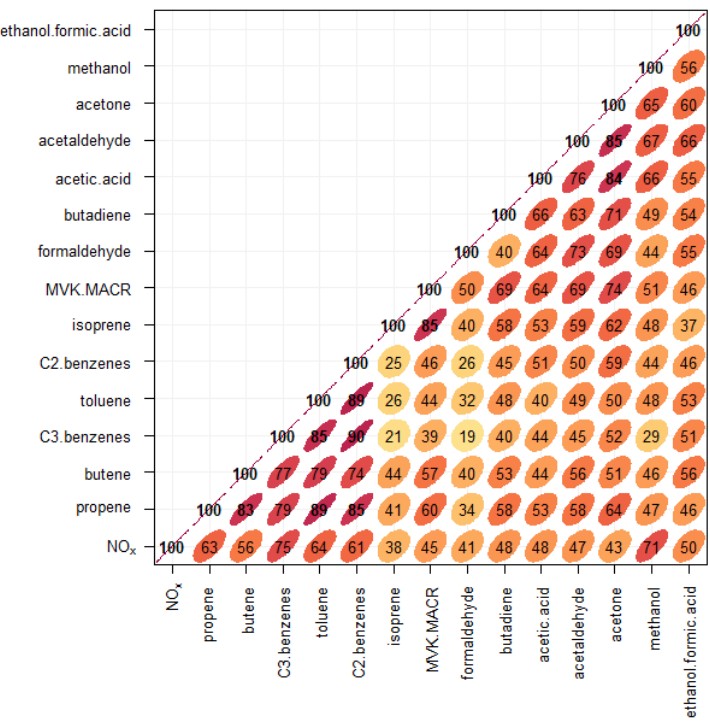

Figure 8. Correlation plot showing the relationship between the fluxes of all identified compounds that contribute more than 0.75% to the total measured VOC flux in the summer measurement campaign as well as the $NO_X$ flux. Numbers and colour represent the % $R^2$ value of each correlation.

**3.4.3 Biogenic fluxes**

Globally, plants are the main source of VOCs into the atmosphere, with isoprene and monoterpenes the main VOC species emitted (Guenther et al., 1995; 2012). However, in urban areas, isoprene is present from both biogenic and vehicular sources (e.g. Borbon et al., 2001; Langford et al., 2010; Valach et al., 2015; Jaimes-Palomera et al., 2016). In the current study, significant BVOC emissions were observed in the summer campaign,

with average emissions of $5.31 \pm 7.69$ nmol $m^{-2}$ $s^{-1}$ for isoprene and $0.21 \pm 0.18$ nmol $m^{-2}$ $s^{-1}$ for monoterpenes. At night-time, emissions of both monoterpenes and isoprene were close to zero with mixing ratios dropping to 0.3 ppbv from a mid-day peak of 0.7 ppbv. Emissions began to increase at 06:00 before peaking at mid-day and returning to zero at around 20:00 (isoprene emissions are displayed in Fig. 6). The summer isoprene flux correlated well ($R^2$ 0.85) with the flux of the isoprene oxidation products methyl vinyl ketone and methacrolein

(MVK+MACR) and poorly with traffic tracers such as toluene, $C_2$-benzenes ($R^2$ 0.26 and 0.25) and $NO_x$ ($R^2$ 0.38) (Fig. 8). This suggests that the contribution from traffic emissions to the summer isoprene flux is low compared to the biogenic source. In the winter campaign, emissions of both isoprene and monoterpenes were comparatively small with average fluxes of $0.67 \pm 3.33$ and $0.06 \pm 0.22$ nmol $m^{-2}$ $s^{-1}$, respectively. Despite the weak isoprene flux, mixing ratios were larger in winter than in summer with mixing ratios consistent through the day at an

average value of 1.21 ppbv. The high isoprene mixing ratio in the winter suggests transport of anthropogenic isoprene from outside of the flux footprint together with a lower rate of photochemical loss than in the summer.

The low winter isoprene flux suggested a small contribution of anthropogenic isoprene to the total flux. Sesquiterpenes were also recorded in the summer with a mean mixing ratio of 0.04 ppbv but a flux was not detected. The summer isoprene flux is larger than that recorded in London (Valach et al., 2015) where an average emission of 0.53 nmol $m^{-2}$ $s^{-1}$ was observed between August and December. This is likely to be due to the London emissions being reduced by the generally lower temperatures and radiation levels, both due to the difference in latitude and also the inclusion of autumn and winter flux data. In the summer campaign predominantly-biogenic compounds accounted for 13% of the total molar VOC flux and 42% of non-oxygenated hydrocarbon emission.

Isoprene emission from biogenic sources is commonly modelled using the Model of Emissions of Gases and Aerosols from Nature (MEGAN, Guenther et al., 2006). Emission of isoprene in MEGAN is calculated as:

$$Emission = \varepsilon \times \gamma \times \rho, \tag{6}$$

where $\varepsilon$ is an emission factor (nmol $m^{-2}$ $s^{-1}$) representative of the isoprene emission from a plant canopy under standard conditions, $\gamma$ is a dimensionless emission activity factor representing deviation from standard conditions and $\rho$ is a factor (normalised ratio) accounting for loss within the canopy. By default, the emission factor used by MEGAN for Beijing is 11.04 nmol $m^{-2}$ $s^{-1}$. The expected isoprene flux from Beijing was modelled using MEGAN together with photosynthetic photon flux density (PPFD) and atmospheric temperature recorded at the IAP metrological tower. The leaf area index (LAI) for Beijing was 1.0 for May and 1.7 in June with these values taken from the MERRA-2 reanalysis data set for 2017. The flux predicted by MEGAN over-estimated the observed flux with the modelled mean diurnal isoprene flux peaking at 21 nmol $m^{-2}$ $s^{-1}$ compared to the measured isoprene flux which peaked at 14 nmol $m^{-2}$ $s^{-1}$ (Fig. 9b).

The measured isoprene flux together with the product of the emission activity factor and loss factor ($\gamma \times \rho$) can be used to optimise the emission factor ($\varepsilon$) for the flux footprint. Langford et al. (2017) used a weighted mean to determine the isoprene emission factor which ensures that the modelled isoprene emission has the same average as the measured flux. This approach gives an emission factor of 10.16 nmol $m^{-2}$ $s^{-1}$ for the summer campaign. Using this emission factor MEGAN estimates a diurnal flux peaking at 18 nmol $m^{-2}$ $s^{-1}$. This value is elevated by the low, anthropogenic, night-time flux observed in Beijing. Alternatively, an optimised isoprene emission factor can be calculated using the least-square regression between the measured isoprene flux and ($\gamma \times \rho$) (Fig. 9a). This gave an isoprene emission factor of 7.57 nmol $m^{-2}$ $s^{-1}$, representing an optimised emission factor for central Beijing. When using this emission factor, the mean diurnal isoprene flux predicted by MEGAN peaked at 14 nmol $m^{-2}$ $s^{-1}$ (Fig. 9b). This compares well with the measured isoprene flux and represents a more appropriate emission factor than that calculated using the weighted mean approach. This optimised emission factor is 69% of the default MEGAN emission factor value for Beijing.

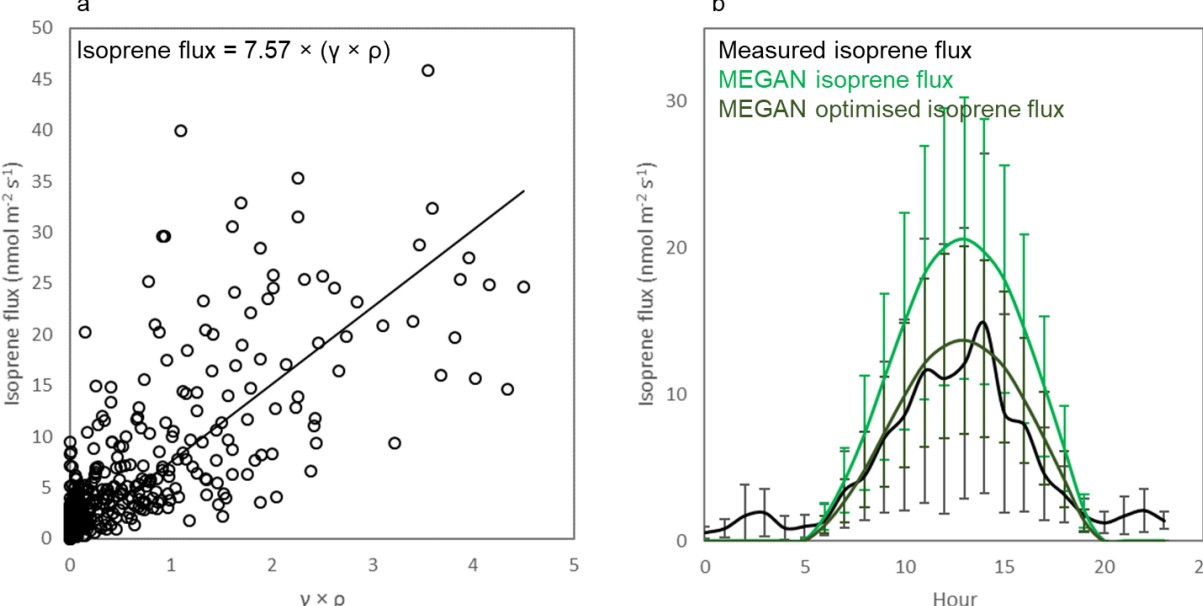

Figure 9. (a) Measured isoprene flux plotted against the product of the emission activity factor ($\gamma$), and the canopy loss and production factor ($\rho$). (b) Diurnal profiles of the isoprene flux predicted by MEGAN using the default and optimised emission factors together with the measured isoprene flux. Error bars represent standard deviation across the measurement period.

### 3.4.4 Impact on atmospheric chemistry

The relative impact of individual VOCs on ozone and peroxyacyl nitrates (PAN) formation can be determined by considering the photochemical ozone creation potentials (POCPs) and the photochemical PAN creation potentials (PPCPs) of the observed compounds. The POCP represents the increment in ground-level ozone production due to the photochemistry of that compound relative to the ozone increment due to ethane. PPCP represents the PAN increment due to the photochemistry of that compound relative to that of propene. The POCPs and PPCPs of 120 organic compounds were reported by Derwent et al. (1998) for UK conditions of the 1990s, and in the absence of more Beijing-specific data these were used to scale both the mixing ratios and the fluxes emitted within the flux footprint of the individual observed VOCs. Derwent et al. (1998) used $NO_x$ concentrations of 0.014 mg m$^{-3}$ as the base case when determining POCPs and tested the POCP calculation at half and double these concentrations. The median $NO_x$ concentration observed in the Beijing summer campaign was 0.0213 mg m$^{-3}$ (Squires et al., 2020), within Derwent's twice base case range. Where a measured mass was not reported by Derwent et al. (1998) the average POCP and PPCP for the appropriate chemical class were applied. It should be noted that as alkanes cannot be detected using PTR-ToF-MS these compounds are not included here. Due to their low POCP and PPCP relative to alkenes and many oxygenated compounds, alkanes are likely to have a limited impact on ozone and PAN formation. However, at large concentrations they can still have a significant impact (Jaimes-Palomera et al., 2016).

Figures 6c and 6d show the concentrations and fluxes scaled by POCP, respectively. Of those compounds resolved by PTR-ToF-MS, predominantly-biogenic compounds were the largest source of POCP making up 34% of the

total POCP-scaled flux. Oxygenated compounds were also a significant source of POCP due to the large molar flux of these compounds, with methanol and ethanol + formic acid making up 13% of the POCP-scaled flux and other oxygenated VOCs 18%. The PPCP-scaled fluxes and concentrations are shown in Figs. 6e and 6f respectively and show that, as with the POCP, oxygenated and predominantly-biogenic VOCs are likely to make the largest contribution to photochemical PAN formation. The large contribution of alkenes, grouped here as alkenes and predominantly biogenic compounds, to ozone formation was also observed in Mexico City where Velasco et al. (2007) found that despite making up 5% of the measured VOC mixing ratio alkenes made the largest contribution to ozone formation.

The influence of VOCs on OH and ozone reactivity was determined by the VOC reaction rate with the OH radical and ozone, relative to that of ethane. VOC reaction rates with the OH radical and ozone were taken from Atkinson and Arey (2003) and references therein. Predominantly-biogenic compounds dominate the OH reactivity scaled VOC flux making up 51% of the potential OH reactivity emitted (Fig. 10b). Oxygenated VOCs and non-aromatic hydrocarbons (predominantly butene and propene) represent 22% and 17% of the VOC flux scaled by OH reactivity respectively. The VOC concentrations scaled by OH reactivity (Fig. 10a) shows that of the VOCs measured using PTR-ToF-MS oxygenated compounds have the largest impact on OH reactivity. This is caused by the very high mixing ratios of these compounds relative to other chemical groups in the atmosphere. However once scaled by OH reactivity predominantly-biogenic compounds, which only make up 4% of the total measured VOC concentration (Fig. 10a), are shown to contribute 21% to the total measured VOCs scaled by OH. Ozone reactivity is dominated by alkenes with predominantly-biogenic compounds and non-aromatic hydrocarbons (predominantly butane, $C_6H_{10}$ and $C_7H_{12}$) making up 31 and 61% of the potential ozone reactivity emitted respectively (Fig. 10d).

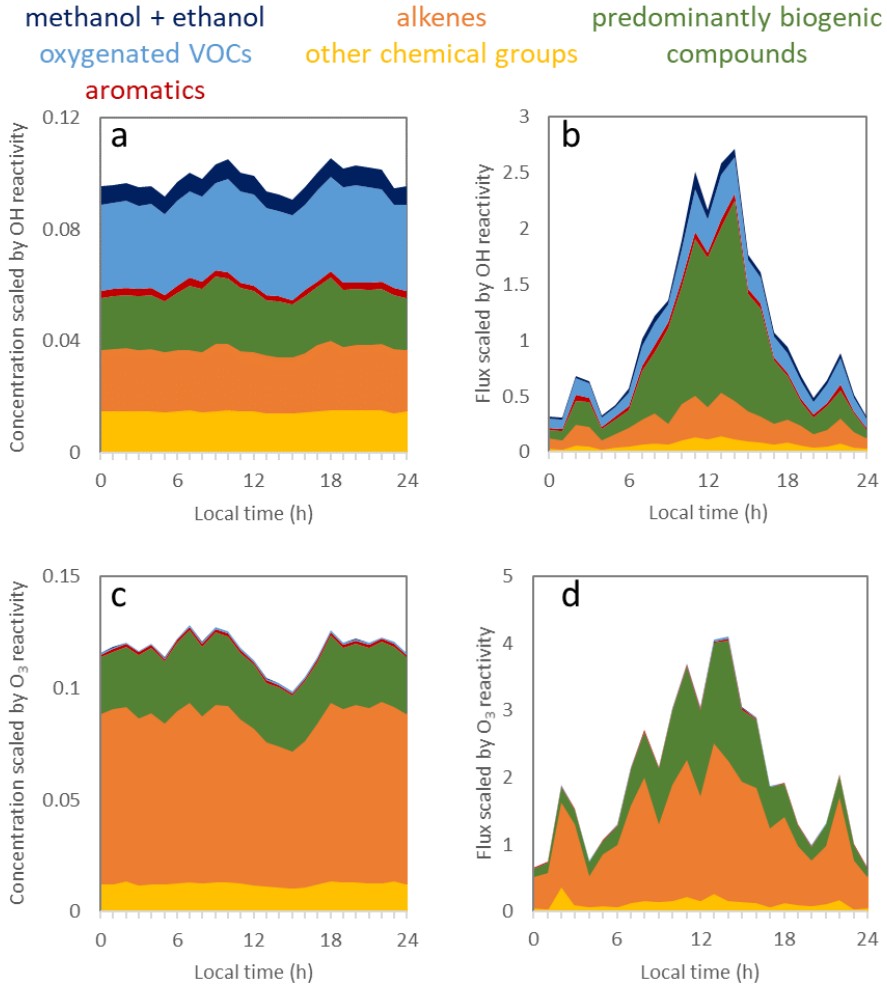

Figure 10. Total observed VOC concentration and flux measured during the summer field campaign scaled using by OH (a, b) and $O_3$ (c, d) reactivity relative to ethane.

**3.5 Emission factor analysis and evaluation**

The Multi-resolution Emissions Inventory for China (MEIC, Li et al., 2017b; Qi et al. 2017) is a comprehensive inventory of the emission of atmospheric pollutants across China and provides the emissions input to many modelling studies (e.g. Hu et al. 2016). Inventories such as MEIC use emissions estimates from individual sources scaled by activity to provide a "bottom-up" emissions estimate. The emission factors used by the inventory are drawn from a broad range of studies (Li et al., 2014; Zheng et al., 2018) with locally derived emission factors used where possible. These emission factors have been summarised by Li et al. (2017c). Whilst activity data is often relatively well constrained at national level, its spatial disaggregation often relying on simplified proxies such as population density adds significant additional uncertainty for the estimate of the emission for a given location. The VOC flux measurements presented here provide a "top-down" measurement at the cityscape-scale which can be used to evaluate the accuracy of the emissions inventory encompassed by the flux footprint. Zhao et al. (2017) assessed the uncertainties in emissions inventories for China and recommended that field studies were used to improve emission estimates.

The MEIC includes emissions from five sectors: power plants, transportation, industrial, agricultural and residential sources with VOC emissions collated by functional group. As no agricultural land was present within the flux footprint, agricultural emissions are not considered here. Monthly emissions data are provided with emissions assumed to be consistent across each month. A diurnal cycle based on local activity data is applied to each emission sector to give a diurnal profile. Within the MEIC inventory VOCs are grouped into classes. Nine of these groupings were considered but those containing alkanes were not considered as alkanes have a proton affinity less than that of water and could not be detected using PTR-MS. The groupings, based on the speciation used by the RADM2 chemical mechanism, are summarised in Table 3. The emission ratio of different VOC species present within each grouping could represent a source of error in the inventory.

VOC emissions estimates for the summer campaign period were calculated through combination of the flux footprint (100 m × 100 m) and high resolution (3 km × 3 km) MEIC v1.3 inventory for 2013. First, the footprint for each flux aggregation period and the inventory grid for the corresponding hour of day were aligned by transforming the flux footprint into the coordinate reference system of the inventory. Subsequently the inventory values were extracted at the centre of each footprint grid cell, creating a pseudo 100 m × 100 m inventory. This was multiplied by the footprint grid, weighting each cell by their contribution to the measured flux. There was little variation between VOC emission factors from the main four emissions inventory cells contributing to the flux observed at the IAP tower site.

Table 3. The measured compounds used to evaluate the accuracy of the MEIC VOC groupings.

| MEIC VOC grouping | Definition | Measured compounds |
|---|---|---|
| HCHO | formaldehyde | formaldehyde |
| ALD | acetaldehyde and higher saturated aldehydes | acetaldehyde; butanal |
| CH3OH | methanol | methanol |
| C2H5OH | ethanol | ethanol + formic acid |
| TOL | toluene and less reactive aromatics | toluene; benzene |
| XYL | xylene and more reactive aromatics | C2 benzenes; C3 benzenes; C4 benzenes; naphthalene; C11H14 |
| KET | ketones | acetone; pentanone |
| ISO | isoprene | isoprene |
| ORA2 | acetic acid and higher acids | acetic acid; propionic acid |

Fig. 11 shows the comparison between measured VOC emission (summer campaign) and the VOC emission predicted by the MEIC inventory for this period with the percentage contribution of each group to the total flux displayed in Fig. 12. Measured emissions of aromatic compounds were 3% and 4% of those predicted by the inventory for TOL and XYL (low mass and high mass aromatics). Squires et al. (2020) compared $NO_x$ and CO fluxes recorded during the APHH-Beijing campaigns with the MEIC inventory and also observed a similar

overestimation of emissions by the inventory, with the inventory overestimating the measured flux 11 and 10 times respectively. Industrial emissions make the largest contribution to aromatic VOC emission in the inventory. However, inspection of the flux footprint reveals few potential industrial sources with the footprint encompassing roads and residential building as well as shops and restaurants. At a coarser resolution of 9 × 9 km the TOL emission peaked at 25 nmol m$^{-2}$ s$^{-1}$, 9 times higher than the measured flux indicating that the overestimate was in part caused by the proxies used to downscale the inventory to 3 km. Previous studies have shown that the inventory underestimates aromatic emission at the Chinese scale (Liu et al., 2012b; Cao et al., 2018), again indicating that the overestimation observed here is a result of the downscaling of the inventory. The allocation of industrial emissions to this residential area is likely a result of downscaling the emissions using proxies such as population (Zheng et al., 2017). Even discounting the industrial emissions, the predicted emissions of TOL and XYL are still 8 times larger than the measured flux at the 3 by 3 km resolution. The inventory does capture the diurnal cycle in emissions with a rapid increase in aromatic VOC emissions at 07:00, emissions remaining high throughout the day before decreasing after 17:00.

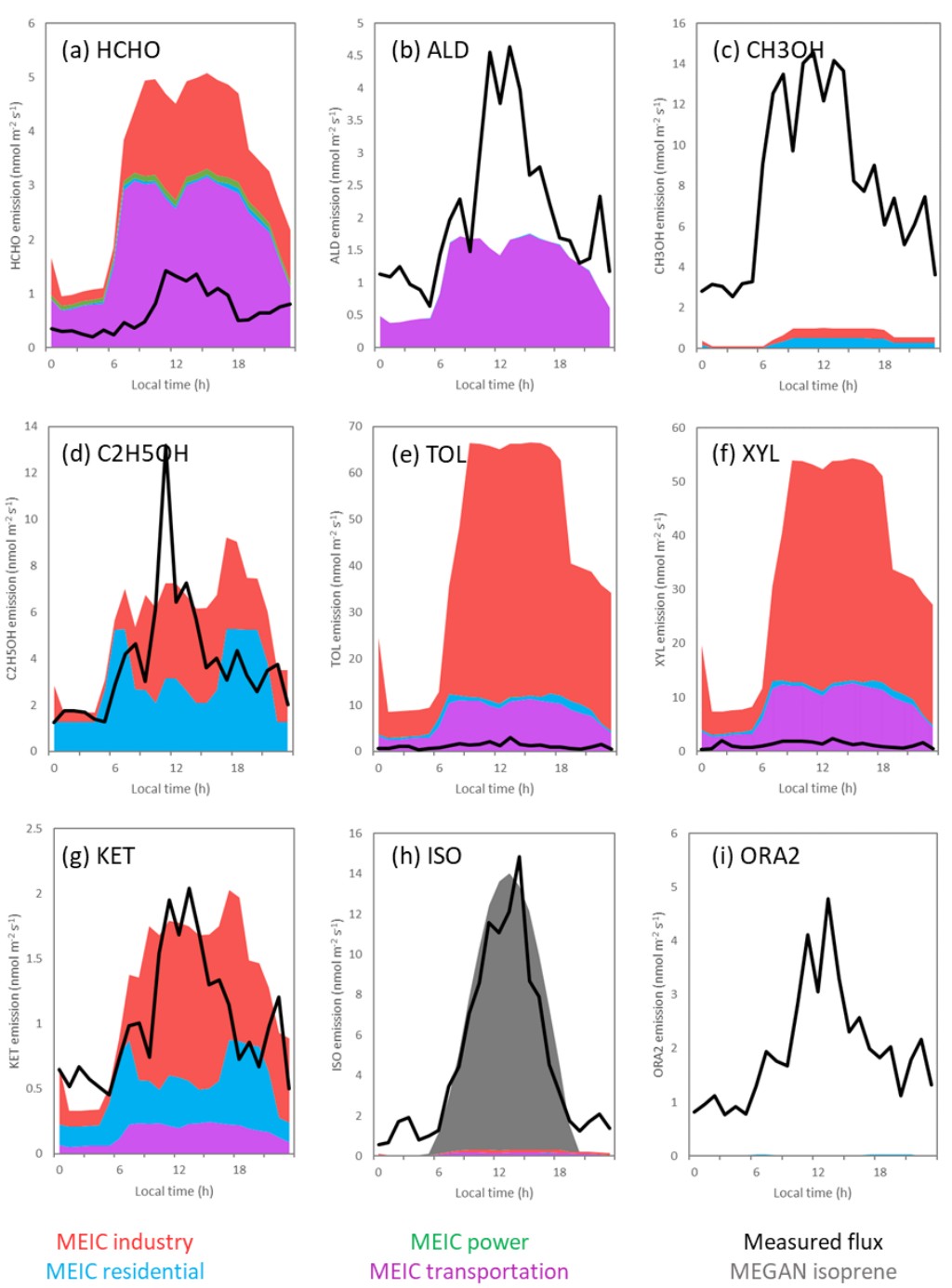

Figure 11. Comparison of 9 VOC classes from the MEIC emissions inventory (Table 3) with measured VOC flux (black line). The diurnal profiles of the isoprene emission predicted by MEGAN using the optimised emission factor is shown in grey (h).

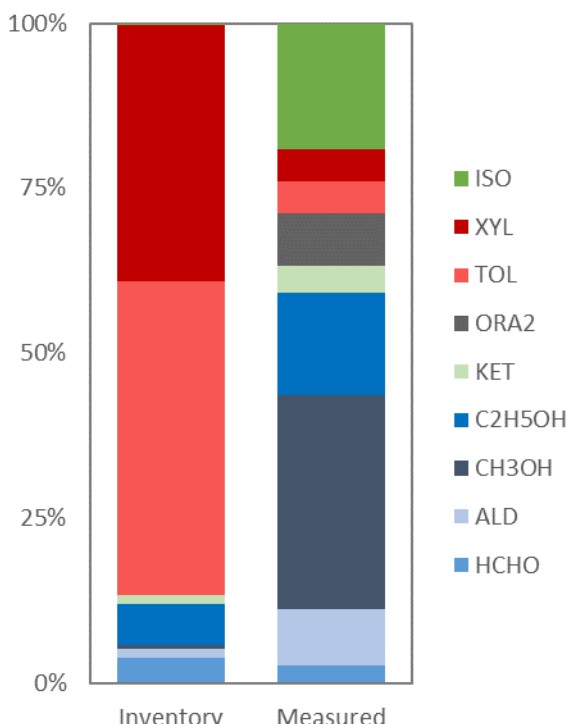

Figure 12. The percentage contribution of the MEIC VOC groupings to the total emission.

Isoprene emissions are significantly underrepresented in the inventory with measured fluxes 20 times higher than inventory emissions. This is expected as the inventory only considers anthropogenic emission and in the summer isoprene emissions are dominated by predominantly-biogenic isoprene. Adding the isoprene emission predicted by MEGAN to the MEIC emission (Fig. 11h) shows that the inclusion of biogenic isoprene enables a more accurate estimate of total isoprene emission. The park at the base of the tower means that a larger biogenic source may be present at the IAP site than would be expected for Beijing on average.

The performance of the inventory in relation to oxygenated VOCs varied considerably between chemical classes. The inventory was more accurate when estimating emissions of ethanol, formaldehyde and ketones with measured emissions 71, 20 and 81% of inventory emissions. When industrial emissions are removed these increase to 132, 30 and 194% of the inventory emissions suggesting that, despite the PTR-ToF-MS only capturing two ketones, ketone emissions are significantly underestimated from residential and transport sources. Measured methanol, aldehydes and organic acid fluxes were also underestimated by the inventory with the measured flux 13, 1.8 and 107 times higher than the inventory, respectively.

It is possible that some underestimation of methanol, aldehydes and organic acid fluxes can be explained by their formation on surfaces following the oxidation of primary-emitted predominantly-biogenic (e.g. Lee et al. 2006; Acton et al., 2018) or anthropogenic compounds (Atkinson, 2000). However, given the ~68 s time taken for an air parcel to reach the inlet point it is not likely that formation in the atmosphere contributes significantly to the flux of these compounds. Consumer goods are known to be a large source of oxygenated VOCs (Dinh et al., 2015; Nematollahi et al., 2019). The use of these products is likely to vary considerably between households, making bottom-up emission hard to estimate.

**4 Summary and conclusions**

Fluxes of speciated VOCs at district scale were measured for the first time in central Beijing with emissions dominated by small oxygenated compounds methanol, acetaldehyde, and ethanol and formic acid which could not be separated. Mixing ratios of most species were significantly higher in the winter than in the summer with VOCs following the "saw tooth pattern" driven by meteorology that has been reported for other pollutants (e.g. Jia et al. 2008). Mixing ratios of aromatic compounds were lower in the summer than in the winter but mixing ratios of small oxygenated VOCs such as methanol were higher, possibly as a result of increased photochemistry but also consistent with their larger emission fluxes.

Stable atmospheric conditions in the winter led to weak turbulent transport making it more difficult to determine their surface emission rate using the micrometeorological eddy covariance method. The observed VOC fluxes in the winter were weaker than those recorded in the summer, probably due to low volatilisation of VOCs and deposition of VOCs transported from outside the city. However, the small data set available for the winter campaign once non-stationary periods are removed prevents firm conclusions from being drawn. The flux in both seasons was dominated by small oxygenated compounds: methanol, acetaldehyde and ethanol + formic acid. These compounds are known to be oxidation products but are also present in many consumer goods products. Fluxes of aromatic compounds in the summer campaign were comparable to those observed over London and Manchester (Langford et al., 2009; Valach et al., 2016; Vaughan et al., 2017) despite the mixing ratios being larger than those observed in those cities. This suggests that the elevated mixing ratios are driven by transport from outside the city. Both fluxes and mixing ratios of toluene were lower than those recorded in Mexico City (Velasco et al., 2009). Comparison of measured VOC fluxes with those predicted by the emissions inventory showed that the inventory failed to capture VOC emission at this local scale. This failure to predict local scale emission is most likely caused by the scaling of city level activity data to the local scale. In order to accurately predict VOC emissions at a local scale higher resolution activity data will be required.

Isoprene and monoterpenes, compounds predominantly emitted from biogenic sources, contributed 13% to the measured molar flux of VOCs but just 3% of the total recorded VOC mixing ratio. Comparatively, oxygenated compounds made up 60% of the molar flux of the compounds resolved by PTR-MS, aromatic compounds made up 7% and other alkenes contributed a further 11%. However, the high reactivity of isoprene and monoterpenes means that their contribution to ozone and PAN formation was greater, with predominantly-biogenic VOCs representing 30 and 28% of the flux contribution to ozone and PAN formation potential, respectively. This effect is even larger in respect to the OH reactivity, where predominantly-biogenic VOCs represent 50% of the total potential OH reactivity of the VOCs emitted and 21% of the mixing ratio when scaled by OH reactivity. Establishing local scale activity data and improved proxies for the scaling of emissions should therefore be a priority when further developing emission inventories for Beijing.

The relatively small emissions of anthropogenic VOC species from central Beijing compared to the large mixing ratios observed suggest that the scope for policy interventions focusing on VOC emission from central Beijing is limited and that the focus must therefore be on emissions controls on all VOC sources, but particularly on vehicles, in regions surrounding the megacity. Predominantly-biogenic compounds make significant contribution to the

photochemical ozone creation potential (POCP), photochemical peroxyacyl nitrates (PAN) creation potential (PPCP) and potential OH reactivity emitted from the city but currently contribute only a small proportion of total reactivity in the atmosphere. However, as transport of VOCs from outside the city is reduced in the future by

785 policy interventions, biogenic sources within the city are likely to become increasingly important to atmospheric chemistry. It is therefore important that emission inventories of VOCs in Beijing should also include an estimate of biogenic VOC emissions, using a tool such as MEGAN with appropriate leaf area index and emission factor values.

**Data availability.** Data are available at https://catalogue.ceda.ac.uk/uuid/7ed9d8a288814b8b85433b0d3fec0300 (last access: 08/04/2020). Specific data are available from the authors on request (wangxm@gig.ac.cn).

**Author contributions**. WJFA made VOC concentration measurements, calculated VOC fluxes and performed VOC data analysis. WJFA prepared the manuscript with contributions from co-authors. ZH, ZW and BD assisted

with VOC concentration measurements. FAS made NOx concentration measurements, calculated their flux and reviewed the manuscript. EN, BL and NM set up instrumentation on the IAP tower, measured wind vector data, provided advice on flux calculations and reviewed the manuscript. WSD, ARV and SM provided support calculating fluxes using eddy4R software and reviewed manuscript. YL performed MEGAN analysis. QZ provided high resolution emissions data. MH processed the raw emissions data into gridded format for comparison

with the measured fluxes. OW assisted with interpretation of the inventory emissions data and provided a detailed review of the manuscript. XW and YZ prepared the PTR-MS instrument and calibration system. PF maintained the tower and site necessary for this work. CER and JL reviewed the manuscript. CNH obtained funding, interpreted data, helped prepare the manuscript and reviewed the manuscript.

**Competing interests**. The authors declare that they have no conflict of interest.

**Acknowledgements**

We thank Lisa Whalley from the University of Leeds for providing photosynthetic photon flux density data. We acknowledge the support from Zifa Wang, Jie Li and Yele Sun from IAP for hosting the APHH-Beijing campaign

at IAP. We thank Zongbo Shi, Di Liu, Roy Harrison, Tuan Vu and Bill Bloss from the University of Birmingham, Siyao Yue, Liangfang Wei, Hong Ren, Qiaorong Xie, Wanyu Zhao, Linjie Li, Ping Li, Shengjie Hou, Qingqing Wang from IAP, Rachel Dunmore and Ally Lewis from the University of York, Kebin He and Xiaoting Cheng from Tsinghua University, and James Allan and Hugh Coe from the University of Manchester for providing logistic and scientific support for the field campaigns. Funding was provided by the UK Natural Environment

Research Council, UK Medical Research Council and the National Science Foundation of China under the framework of the Newton Innovation Fund (grant NE/N006976/1 to Lancaster University and grant 41571130031 to the Guangzhou Institute of Geochemistry). A full list of other grants that directly and indirectly supported this

work is given in Shi et al. (2019). The National Ecological Observatory Network is a project sponsored by the National Science Foundation and managed under cooperative agreement by Battelle. This material is based upon work supported by the National Science Foundation (Grant DBI-0752017). Any opinions, findings, and conclusions or recommendations expressed in this material are those of the author and do not necessarily reflect the views of the National Science Foundation.

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
