# Peer review of "Surface-atmosphere fluxes of volatile organic compounds in Beijing"

_Atmospheric Chemistry and Physics, 2020_

## Referee Comment (RC1) · Erik Velasco (Referee) · 26 May 2020

**Comments on 'Surface-atmosphere fluxes of volatile organic compounds in Beijing' by Acton et al.**

This manuscript presents eddy covariance flux measurements of volatile organic compounds (VOCs) over a district of Beijing, China. The eddy covariance method has been widely used to measure urban fluxes of carbon dioxide ($CO_2$) during the last 1-2 decades, but only a handful of studies has measured fluxes of VOCs. In this context, the material presented in this manuscript gains relevance and should contribute to improve current knowledge on emission patterns of precursor species, as well as provides unique data to evaluate the accuracy of gridded emission inventories. However, before the manuscript can be considered for further revision and potential publication, the authors need to address the following issues. Details are provided in the specific comments.

- The authors need to explain the advantages and disadvantages of the eddy covariance method with respect to other methods to evaluate urban emissions. The conditions in which eddy covariance flux towers can be installed need a close revision. A full understanding of the eddy covariance assumptions is also needed.

- The facilities used in this study to deploy the eddy covariance system to measure fluxes of VOCs have been previously used to measure fluxes of $CO_2$ (e.g., Song et al., 2013; Liu et al., 2012; Song and Wang, 2012). A review of such studies is recommended.

- Previous articles describing VOC flux measurements by eddy covariance in Mexico City (Velasco et al., 2005, 2009) and Innsbruck (Karl et al., 2018) could be used as references to analyze and present the results of this study. The fluxes reported for Mexico City provide valuable information to compare the fluxes observed in Beijing considering that both are large cities of developing nations. Similarly, the way Karl et al. (2018) analyzed and presented a large number of VOC species could be followed.

- The discussion is biased to VOC studies in cities from UK. The manuscript will be strengthened if the findings are compared to results of previous studies conducted in China and other large cities. For example, the relationship between mixing ratios of selected VOC species has been widely used as a mean to evaluate the characteristics of urban emissions (e.g., Parrish et al., 2009; von Schneidemesser et al., 2010; Velasco et al., 2007).

- The authors need to demonstrate that the eddy covariance set up faithfully captures the entire range of energy-carrying eddies through inspection of the (co)spectra of the measured variables, and therefore is capable of measuring meaningful and representative turbulent fluxes. Similarly, a comprehensive description of the monitored footprint is needed to explain the observed fluxes.

- The writing needs some improvement.

**Specific comments (page, line)**

**2,35.** The abstract can be shorter. The first paragraph provides generic information. For example, the severe air pollution problem of Beijing is well known, and there is no need to specify in which institute the measurements were conducted.

**2,51-52.** In which aspects, magnitude or/and temporal distribution?

**2, 68-69.** Urban emissions have been widely evaluated through a number of direct and indirect methods. A brief discussion on the emissions data provided by those methods in comparison to the data obtained from eddy covariance flux towers will put in context the information presented in this study.

**3, 72-74.** Be more specific, vehicular exhaust no. 1 and vehicular exhaust no. 2 do not provide major information.

**3, 78-79.** One of the main goals of the quality monitoring is to evaluate the effectiveness of the control measures in place. Many cities around the world have made important investments to build air quality monitoring networks to measure and report hourly concentrations of key airborne pollutants, including particulate matter and speciated VOCs.

**3, 93.** Activity data?

**3, 95-97.** The application of the eddy covariance method to evaluate the accuracy of gridded emission inventories of VOCs in urban areas was done by first time in Mexico City (see Velasco et al., 2005, 2009).

**3, 99-106.** A statement of the hypothesis to test including the study objectives in context of the APHH project will help to analyze the flux observations and reach strong conclusions.

**4, 100-119.** Why was the AIP site selected to deploy an eddy covariance system? Does it meet the eddy covariance assumptions of homogeneity in terms of land cover and emissions distribution? A much more comprehensive description of the monitored site is needed. Information of the land cover distribution, urban morphology, roughness elements, trees characteristics, vehicular traffic, population density, etc. is needed. Please, consider the Local Climate Zone classification (see Stewart and Oke, 2012).

**4, 117-118.** Urban background in terms of ambient concentrations or urban fluxes?

**4, 126.** '… at a height of 102 m above ground level' was mentioned in the previous paragraph.

**5,155.** The legend should guide readers to understand the sketch of the sampling system presented in this figure without need of going back to the text.

**5, 162-163.** You may consider moving this table to the main article since it is the only material presented as Supplementary Information.

**7, 200.** The pioneering studies in Mexico City mentioned in a previous comment were done using the disjunct eddy covariance method and the original PTR-MS instrument.

**7, 201.** A detailed description of the flux data postprocessing? The eddy covariance method does not estimate fluxes, does measure fluxes.

**7, 203-204.** Turbulent fluxes are computed using instantaneous deviations or fluctuations of the vertical wind velocity and VOC mass density (e.g., mg m$^{-3}$).

**7, 211-212.** What about the time taken by the instrument to analyze the sample?

**7, 228-229.** Why did Squires et al. (2020) choose this method?

**7, 240-241.** Do you mean stationarity?

**7, 242.** Explain briefly the stationarity test.

**7, 245-247.** There are methodologies available to fill gaps in time series of turbulent fluxes measured by eddy covariance. However, depending on the study purposes it may not be necessary to fill such gaps. Consider that in urban environments VOC fluxes respond mainly to anthropogenic and not natural variables like in a forest. Human actions can be random at times (e.g., traffic jams), thus very low or very high fluxes are possible.

**7, 238-249.** Please, indicate the number of periods excluded for further analysis due to lack of stationarity and/or enough turbulence. How many averaging periods were affected by rain/snow and instrument maintenance?

**8, 254-255.** This information was already provided.

**9, 263-264.** This figure and the description of the dominant winds during the field observations are useless without a proper analysis of the land cover and distribution of the measured fluxes by wind sectors.

**9, 265-266.** This information was already provided.

**10, 279-281.** Is this true at nighttime and daytime? Atmospheric stability evolves during the diurnal cycle, thus significant changes in the footprint can be expected.

**11, 292-296.** This figure is incomplete. A color gradient to indicate the flux contribution is missing. No spatial scale was included neither a north indicator. The article is in English, but the map includes text in Chinese. The legend refers to a grid of cells, but no cells are included. Do these footprints correspond to nighttime or daytime?

**11, 302-303.** Define 'sawtooth cycle'. Although the paper is about fluxes, it will be good to provide a short description and a figure about their diurnal cycle.

**12, 322-330.** This paragraph should be part of the methodology. Do not repeat information.

**13, 349-351.** Turbulent fluxes measured by eddy covariance represent the VOC exchange within the observed footprint. Periods affected by advection were removed through the stationarity criterion, isn't it?

**13, 351-352.** This is unclear. Do you mean that turbulent fluxes cannot be measured during winter in Beijing, and therefore half of the study was useless?

**14, 366.** Table 2. A complete list of the 50 VOC species identified and evaluated in this study should be included in the Supplementary Material. You may use as reference how Karl et al. (2018) analyzed and presented an even larger list of VOC species.

**14, 378-382.** This section discusses fluxes no mixing ratios. Mixing ratios were discussed in section 3.3.

**14, 380-382.** London and Manchester are only two cities. There are many papers reporting mixing ratios of oxygenated VOCs for cities of China and the rest of the world.

**15, 387.** Vehicular traffic is a major source of isoprene in urban areas (e.g., Borbon et al., 2001; Jaimes-Palomera et al., 2016).

**17, 400.** Do you mean less than 1%? Figure 6 has not been cited yet in the text.

**17, 403.** Be more specific on the type of combustion sources (e.g., vehicular traffic, industry, biomass burning).

**17, 409.** A vehicular emission control program?

**17, 412-415.** A comparison against the fluxes reported by Velasco et al. (2005, 2009) for two locations of Mexico City will help to put in context the fluxes observed in Beijing with respect to other large city of a developing nation.

**17, 416-418.** How do these ratios between mixing ratios and fluxes compare to those reported in the literature for other cities?

**17, 420-425.** Propene is dominated by emissions from vehicular traffic in locations with heavy traffic (see Velasco et al, 2005, 2009). Averaging periods affected by large plumes are expected to be removed by the stationarity criterion. If stacks of large industries were within the monitored footprint and their impact was not filtered by the stationarity criterion, the turbulent fluxes reported in this study are suspicious.

**17, 430-432.** See Fig. 4 in Velasco et al. (2005) and Fig. 3 in Velasco et al. (2009).

**19, 445.** The acronyms POCP and PPCP have not been defined in the text yet.

**20, 446-449.** A similar figure showing the flux correlations between species during the winter period is needed.

**20, 466-467.** See previous comment on the vehicular contribution to urban isoprene.

**21, 481-482.** How and who determined this emission factor for Beijing's urban vegetation?

**22, 511-514.** Based on the comprehensive set of measurements conducted during the APHH study, is it possible to adjust the POCP and PPCP factors for Beijing's conditions? If it is not possible, wouldn't be better to evaluate only the OH reactivity of each compound?

**22, 514-516.** Light alkanes can be important players if their emissions are high like in Mexico City (see Jaimes-Palomera et al., 2016).

**24, 540.** The VOC classification used here is 'strange'. Be more specific on the species included in the group 'other hydrocarbons'.

**24, 543.** Consider a different title for this section. Why did the evaluation of the accuracy of the gridded emissions inventory was limited to the summer period?

**24, 544-546.** This was already mentioned in the introduction (lines 90-94). Indicate the emissions inventory's year reference.

**25, 553-554.** Were any power plant or large industry within the monitored footprint? If the answer is positive, the fluxes reported here might not be valid. Review the basic assumptions for measuring turbulent fluxes by eddy covariance (Velasco and Roth, 2010).

**25, 575-576.** The poor agreement between measured fluxes and estimated emissions might respond to the size of the grids in the emissions inventory. Cells of 9 km$^2$ might be too large to capture the emissions observed by the flux tower.

**27, 590.** According to the industrial emissions reported in the inventory developed for air quality modelling and presented in this figure, the site selected to conduct flux measurements by eddy covariance in Beijing is not adequate. Review the conditions needed to perform representative eddy covariance flux measurements in urban areas (Velasco and Roth, 2010).

**28, 597-599**. If it was known in advance that the inventory did not include the biogenic component, then which was the purpose of comparing the measured fluxes of isoprene? The emissions estimated by Megan should be added to those reported by the MEIC inventory.

**28, 608-610.** Indeed, the presence of methanol, aldehydes and organic acids can be explained by local atmospheric chemistry, however, it was previously proved that the time taken by an air parcel to reach the top of the tower was much less than the time needed for the oxidation of such compounds. If it was not the case, fluxes of those compounds cannot be measured by eddy covariance in Beijing.

**29, 617.** Instead of presenting conclusions and final remarks, this section is a summary of results.

**29, 618.** Rewrite this sentence. For example: 'Fluxes of speciated VOCs at district scale were measured by first time in Beijing ….'.

**29, 625.** 'Stable atmospheric conditions' may sound better.

**29, 626-627.** It may be true for VOC species emitted by evaporation of fossil fuels and solvents, but not for species emitted by other emission sources. It is not possible to talk about advection of VOCs in the case of fluxes measured at district level by eddy covariance, see previous comments.

**29, 630-633.** See previous comments about comparing the fluxes reported in this study with those observed in Mexico City.

**References**

Borbon, A., Fontaine, H., Veillerot, M., Locoge, N., Galloo, J. C., Guillermo, R.: An investigation into the traffic-related fraction of isoprene at an urban location, Atmos. Environ. 35, 3749–3760, 2001.

Jaimes-Palomera, M., Retama, A., Elias-Castro, G., Neria-Hernández, A., Rivera-Hernández, O. and Velasco, E.: Non-methane hydrocarbons in the atmosphere of Mexico City: Results of the 2012 ozone-season campaign. *Atmospheric environment*, *132*, 258-275, 2016.

Karl, T., Striednig, M., Graus, M., Hammerle, A., Wohlfahrt, G.: Urban flux measurements reveal a large pool of oxygenated volatile organic compound emissions. *Proceedings of the National Academy of Sciences*, *115*(6), 1186-1191, 2018.

Liu, H.Z., Feng, J.W., Jarvi, L. and Vesala, T.: Four-year (2006-2009) eddy covariance measurements of $CO_2$ flux over an urban area in Beijing. *Atmospheric Chemistry & Physics*, *12*(17), 2012.

Song, T. and Wang, Y. Carbon dioxide fluxes from an urban area in Beijing. *Atmospheric Research 106*, 139-149, 2012.

Song, T., Sun, Y. and Wang, Y.: Multilevel measurements of fluxes and turbulence over an urban landscape in Beijing. *Tellus B: Chemical and Physical Meteorology*, *65*(1), 20421, 2013.

Stewart I.D., Oke, T.R.: Local climate zones for urban temperature studies. *Bull. Am. Meteorol. Soc.* 93, 1879–1900, 2012.

Velasco, E., Lamb, B., Pressley, S., Allwine, E., Westberg, H., Jobson, B. T., Alexander, M., Prazeller, P., Molina, L., Molina, M.: Flux measurements of volatile organic compounds from an urban landscape, *Geophys. Res. Lett*., 32, L20802, 2005.

Velasco, E., Lamb, B., Westeberg, H., Allwine, E., Sosa, G., Arriaga-Colina, J. L., Jonson, B. T., Alexander, M. L., Prazeller, P., Knighton, W. B., Rogers, T. M., Grutter, M., Herndon, S. C., Kolb, C. E., Zavala, M., de Foy, B., Volkamer, R., Molina, L. T., and Molina, M. J.: Distribution, magnitudes, reactivities, ratios and diurnal patterns of volatile organic compounds in the Valley of Mexico during the MCMA 2002 & 2003 field campaigns, *Atmos. Chem. Phys*., 7, 329–353, 2007.

Velasco, E., Pressley, S., Grivicke, R., Allwine, E., Coons, T., Foster, W., Jobson, B.T., Westberg, H., Ramos, R., Hernández, F., Molina, L.T.: Eddy covariance flux measurements of pollutant gases in urban Mexico City. *Atmos. Chem. Phys.* 9, 7325–7342, 2009.

Velasco, E., Roth, M.: Cities as net sources of $CO_2$: review of atmospheric $CO_2$ exchange in urban environments measured by eddy covariance technique. *Geogr. Compass* 4, 1238–1259, 2010.

Parrish, D.D., Kuster, W.C., Shao, M., Yokouchi, Y., Kondo, Y., Goldan, P.D., de Gouw, J.A., Koike, M. and Shirai, T.: Comparison of air pollutant emissions among mega-cities. *Atmospheric Environment*, *43*(40), 6435-6441, 2009.

von Schneidemesser, E., Monks, P.S. and Plass-Duelmer, C.: Global comparison of VOC and CO observations in urban areas. *Atmospheric Environment*, *44*(39), 5053-5064, 2010.

---

## Referee Comment (RC2) · Anonymous Referee #2 · 26 May 2020

This manuscript presents the VOC concentration and flux measurements using a PTR-MS at a tower in Beijing during the China-UK APHH summer and winter campaigns. Large concentrations and fluxes of small OVOCs were measured in both winter and summer. The measurements also show high concentrations of isoprene and aromatics in both summer and winter, although the surface-to-air fluxes were small. The authors compared the results to the fluxes predicted by the MEIC (anthropogenic) and MEGAN (biogenic) emission inventories and attempted to explain some of the discrepancies.

The topic of the manuscript is of interest to the community and certainly within the scope of ACP. The measurement methodology and analytical methods are standard. Most of the results appear reasonable. However, the interpretation of the isoprene and aromatic sources have some contradictions with previous studies and the known

characteristics of the urban site, warranting more in-depth analysis. Also, the writing has a fairly large number of errors in grammar and punctuation usage. The senstences were often too long and commas are often missing. I therefore recommend that the paper undergo major revision to address some of the key issues and also to improve the writing.

Major comments:

Figure 3a: Why was the timeseries of the meterological variables drawn for November 16 to December 7 for the winter campaign? This does not match the duration of the winter campaign nore that of the PTR-MS measurements stated in lines 254-255.

Lines 308-309: "In addition, isoprene, which has both biogenic and anthropogenic sources, had a median mixing ratios of 1.08 and 0.38 ppb in the winter and summer campaigns respectively.": This is an interesting result. Although several studies have indicated possible anthropogenic emissions of isoprene (potentially related to traffic), the origin of this anthropogenic isoprene have not been fully explored. Most researchers still predominantly associate isoprene with biogenic emissions. Also, I do not think I have ever seen isoprene concentrations this much higher in winter (median 1.08) than in summer (median 0.38). Could the authors elaborate a little on this anthropogenic isoprene? Were there strong correlations between isoprene concentrations or fluxes with other source-specific VOCs? Figure 5 shows that there was not a strong positive flux of isoprene in winter. This contradicts (1) the findings from previous studies suggesting that the anthropogenic isoprene is traffic-related, and (2) the fact that the IAP tower is located near major roads with heavy traffic (major ring roads within its 1-2 km flux footprint). Also, it would be helpful to cite some key reference on the anthropogenic isoprene source in lines 308-309.

Lines 346-348: "The weak VOC flux observed in the winter may, in part, be caused by low volatilisation of VOCs in winter conditions, where the average temperature was 3.6 °C compared to 25 °C in the summer.": But most of the the species shown in Fig 5 and

Table 2 should be sufficiently volatile even at 3.6oC (e.g., methanol, ethanol, acetaldehyde, etc). I think, to be more illuminating, the authors should state the known major sources of these low-molecular-weight organics and indicate whether those sources are smaller in winter.

Lines 380-384: What seasons were these measurements conducted?

Lines 386-388: If the anthropogenic contribution of isoprene is so significant at this site in winter (especially given the much larger concentration in winter than in summer), how can one catagorically label isoprene as biogenic, even in summer?

Lines 399-401: "The relationship between the fluxes of VOC species which contribute more than 0.75% to the total measured VOC flux and 400 the NOx flux in the summer measurement period is shown in Fig. 7.": This sentence is grammatically incorrect and therefore impossible to understand. Please kindly revise.

Lines 402-403: "NOx in urban areas is a combustion product so the NOx flux is used here as proxy for combustion sources of pollutants": At this site, the NOx flux is most strongly a proxy for vehicular emissions.

Lines 426-428: "The close correlation with the NOx flux and aromatics such as toluene suggests a combustion source contributes to the total methanol flux but the summer methanol flux also correlates well with small oxygenated VOCs such as acetaldehyde.": What is the evidence for the first half of this statement? Particularly since it contradicts the second half of the sentence.

Section 3.4.3: Again, I think the source of isoprene at this IAP site should be more carefully interpreted, given that a large concentation was observed in winter.

Lines 468-470: "The low winter isoprene flux suggested a small contribution of anthropogenic isoprene to the total flux.": I am not convinced. Previous studies indicate that the anthropogenic emission of isoprene was vehicular. And this IAP tower site should be strongly affected by traffic emissions.

Lines 485-488: "The flux predicted by MEGAN over-estimated the observed flux with the modelled mean diurnal isoprene flux peaking at 21 nmol m-2 s-1 compared to the measured isoprene flux which peaked at 14 nmol m-2 s-1 (Fig. 8b).": How does the measured flux in summer compare with the top-down estimates using satellite-based observations by Cao et al. (2018)?

Lines 512-514: "The POCPs and PPCPs of 120 organic compounds were reported by Derwent et al. (1998) for UK conditions of the 1990s, and in the absence of more Beijing-specific data these were used to scale both the mixing ratios and the fluxes emitted within the flux footprint of the individual observed VOCs." What were the NOx levels used in the Derwent et al. (1998) calculation, and how different was it with the present-day Beijing NOx levels? Also, how long were the POCPs and PPCPs calculated for in Derwent et al. (1998), i.e, were these ozone produced over a day or several days? This makes a difference because methanol and other VOCs measured in abundance in this campaign have lifetimes > 1 day. Using 1-day POCPs and PPCPs would estimate the ozone and PAN production potentials on a local scale.

Lines 547-549: "Whilst activity data is often relatively well constrained at national level, its spatial disaggregation often relying on simplified proxies such as population density, adds significant additional uncertainty for the estimate of the emission for a given lo-cation.": The emission ratios (i.e, source species profile) for different NMVOC species may present an even larger uncertainty to the final inventory.

Lines 575: "Measured emissions of aromatic compounds were 3% and 4% of those predicted by the inventory for TOL and XYL (low mass and high mass aromatics).": This is very surprising, unless the high-resolution version of MEIC placed a disproportionally large emission near the IAP site. Several previous studies have indicated at least some underestimation of Chinese aromatic emissions in MEIC or its predecessor inventory (Liu et al., 2012; Cao et al. 2018).

Lines 579-580: "Industrial emissions make the largest contribution to aromatic VOC

emission in the inventory.": Is this statement drawn from the numbers for the local grids, or from the national inventory? The authors measurements clearly show that the aromatics were traffic-related, and I think the same was true in the MEIC inventory.

Section 3.5: There has been several studies using formaldehyde and glyoxal observed by satellite instruments to constrain NMVOC emissions in a top-down fashion. While the resolutions were very different from those here, it might still be helpful to compare their results with the measurements here.

Lines 646-649: "The relatively small emissions of anthropogenic VOC species from central Beijing compared to the large mixing ratios observed suggest that the scope for policy interventions focusing on VOC emission from central Beijing is limited and that the focus must therefore be on emissions controls in regions surrounding the megacity.": Is this conclusion consistent with the observed high toluene emission and its high correlation with NOx (traffic)?

Minor comments:

Line 129-130: Is it possible that there is a negative bias of semi-volatile organics by filtering PM2.5?

Line 233: "Loss of high frequency flux was due to measuring at 5 Hz was estimated to be less than 10%.": missing 'and' between '5 Hz' and 'was'.

Line 234: "Squires et al. (2020) estimated that the average the time taken for an air parcel ...": remove 'the' before 'time'

Line 238: "quality-assessed"

Line 242-244: I do not understand why the quality control for flux was done on a 'file' basis. How long is the record in each file? Shouldn't the quality control be done on a certain timescale?

Lines 348-349: "In the winter the positive (emission) flux of VOCs from the city is likely

suppressed by deposition of VOCs transported at, high concentrations, from outside of the city.": This sentence is unclear; maybe because the first comma is misplaced?

Lines 349-351: "While most mean winter VOC fluxes remained positive this term will be a balance between emission from the city and deposition of VOCs transported into the footprint region.": I do not understand this sentence. Please kindly clarify.

Line 372: Missing comma after "In the summer campaign"

LIne 377: "can't" should be cannot

Reference: Cao, H., T.-M. Fu, L. Zhang, D. K. Henze, C. Chan Miller, C. Lerot, G. Gonzalex Abad, I. De Smedt, Q. Zhang, M. van Roosendael, K. Chance, J. Li, J. Zheng, and Y. Zhao (2018), Adjoint inversion of Chinese non-methane volatile organic compound emissions using space-based observations of formaldehyde and glyoxal, Atmospheric Chemistry and Physics, 18, 15017-15046, doi:10.5194/acp-18-15017-2018.

Liu, Z., Wang, Y., Vrekoussis, M., Richter, A., Wittrock, F., Burrows, J. P., Shao, M., Chang, C.-C., Liu, S.-C., Wang, H., and Chen, C.: Exploring the missing source of glyoxal (CHOCHO) over China, Geophys. Res. Lett., 39, L10812, https://doi.org/10.1029/2012gl051645, 2012.
* * *

---

## Author Comment (AC1) · 29 Jul 2020

This document includes the authors' responses to both reviewer 1 and reviewer 2. Reviewers' comments are in bold text and the authors' responses are in plain text. Text taken from the manuscript is presented in quotation marks. The updated manuscript is presented below with changes to the text highlighted in yellow.

**Response to RC1: Erik Velasco**

**This manuscript presents eddy covariance flux measurements of volatile organic compounds (VOCs) over a district of Beijing, China. The eddy covariance method has been widely used to measure urban fluxes of carbon dioxide (CO2) during the last 1-2 decades, but only a handful of studies has measured fluxes of VOCs. In this context, the material presented in this manuscript gains relevance and should contribute to improve current knowledge on emission patterns of precursor species, as well as provides unique data to evaluate the accuracy of gridded emission inventories. However, before the manuscript can be considered for further revision and potential publication, the authors need to address the following issues. Details are provided in the specific comments.**

Response: The authors thank the reviewer for their constructive comments and for taking time to review the manuscript, responses to specific comments are set out below.

**• The authors need to explain the advantages and disadvantages of the eddy covariance method with respect to other methods to evaluate urban emissions. The conditions in which eddy covariance flux towers can be installed need a close revision. A full understanding of the eddy covariance assumptions is also needed.**

Response: The introduction has been expanded to give a description of the advantages of eddy covariance flux measurement and to highlight previous flux measurements made in urban environments. The site description has been expanded to include discussion of the flux inlet location relative to the height of surrounding buildings.

Introduction:

"Until comparatively recently it was very difficult to validate VOC emissions inventories at the cityscape scale, but the development of the PTR-TOF technology now allows VOCs to be measured with sufficient time resolution for flux estimates to be made using the method of eddy covariance based on simultaneous micrometeorological measurements. Although the conditions under which eddy covariance can be applied in the heterogeneous city environment are somewhat limited by flow interferences from neighbouring buildings, night-time boundary layer stability and other considerations, the method has been successfully used for estimating both $CO_2$ fluxes (e.g. Song et al., 2013; Liu et al., 2012a; Song and Wang, 2012) and VOC fluxes (e.g. Velasco et al., 2005; 2009; Langford et al., 2009; Valach et al., 2015; Vaughan et al., 2017; Karl et al. 2018). However, VOC flux measurements have not previously been attempted in Beijing."

Site description:

"The IAP mast was used for the flux measurements as it allowed the sampling inlet and sonic anemometer to be positioned in the constant flux layer. The flux inlet was located at 102 m, this is over twice the mean height (45 m) of the tall residential building to the southwest (Liu et al., 2012a) and below the mixing layer height (Squires et al., 2020)."

**• The facilities used in this study to deploy the eddy covariance system to measure fluxes of VOCs have been previously used to measure fluxes of CO2 (e.g., Song et al., 2013; Liu et al., 2012; Song and Wang, 2012). A review of such studies is recommended.**

Response: A short discussion of the $CO_2$ fluxes made at this site has been added to section 3.4.

"Fluxes of $CO_2$ have been previously recorded at the IAP field site (Liu et al., 2012; Song and Wang, 2012; Song et al., 2013). Liu et al. (2012a) reported four years of $CO_2$ eddy covariance measurements showing a seasonal variation in emissions driven by winter heating and uptake of $CO_2$ by vegetation in the summer. Spatial variation in emission was shown to be determined by the surface cover and that the diurnal profile of emissions was largely dependent on traffic volumes."

**• Previous articles describing VOC flux measurements by eddy covariance in Mexico City (Velasco et al., 2005, 2009) and Innsbruck (Karl et al., 2018) could be used as references to analyze and present the results of this study. The fluxes reported for Mexico City provide valuable information to compare the fluxes observed in Beijing considering that both are large cities of developing nations. Similarly, the way Karl et al. (2018) analysed and presented a large number of VOC species could be followed.**

Response: Discussion of the VOC fluxes has been expanded to include comparison with the measurements made in Mexico City. Reference to Karl et al. (2018) was included in the discussion (line 384 of the ACPD version). The PTR-QiTOF used by Karl et al. (2018) allows a 10-fold decrease in the flux detection limit when compared to first generation PTR-ToF-MS instruments such as the PTR-TOF-MS 2000 used in this study. Therefore, while Karl et al. detected ~300 ions which were then included in their flux analysis, in this study we identified 65 masses with a median relative random error of less than 150%. Due to this limitation, we focused this study on identifying the dominant VOC fluxes and understanding their impact on atmospheric chemistry as opposed to the detailed analysis of all masses provided by Karl et al. (2018).

**• The discussion is biased to VOC studies in cities from UK. The manuscript will be strengthened if the findings are compared to results of previous studies conducted in China and other large cities. For example, the relationship between mixing ratios of selected VOC species has been widely used as a mean to evaluate the characteristics of urban emissions (e.g., Parrish et al., 2009; von Schneidemesser et al., 2010; Velasco et al., 2007).**

Response: The discussion is limited by the manuscript's focus on VOC flux measurements, with VOC mixing ratios described only to provide context to the fluxes. There are a limited number of studies reporting VOC fluxes in urban areas and we thank the reviewer for drawing our attention to the pioneering studies in Mexico City (see previous comment). Parrish et al. (2009) and von Schneidemesser et al. (2010) have been added to the introduction to provide additional context to this study. Reference to the influence of different VOC chemical groups on ozone production in Mexico City (Velasco et al., 2007) has been added to section 3.4.4. As suggested, we have expanded the discussion of VOC mixing ratios in section 3.3 to make reference to studies conducted in China.

**• The authors need to demonstrate that the eddy covariance set up faithfully captures the entire range of energy-carrying eddies through inspection of the (co)spectra of the measured variables, and therefore is capable of measuring meaningful and representative turbulent fluxes. Similarly, a comprehensive description of the monitored footprint is needed to explain the observed fluxes.**

Response: Many thanks for the reviewer's suggestions, which we used as an opportunity to provide additional technical information in the manuscript. We updated section 2.5: Flux calculation to read:

"This loss of low frequency flux was investigated by Squires et al. (2020) who found that at the 102 m measurement height the flux loss was ca. 7%. Due to similarity of the transport mechanisms underlying the low frequency flux loss (e.g., Mauder et al., 2020) we assume this estimate to apply also to the simultaneously measured VOC fluxes presented here. We did not correct the fluxes presented in this paper for low frequency flux loss. We further investigated the effect of high-frequency spectral loss on the VOC fluxes using a wavelet-based methodology (Nordbo and Katul, 2013). Loss of high frequency flux due to measuring at 5 Hz was estimated to be less than 10%, which we did not correct for in the presented fluxes."

The reviewer points out the importance of "measuring meaningful and representative turbulent fluxes", to which we agree wholeheartedly. We would like to point out that in addition to flux spectral losses and footprint, flux data quality considerations are also important to achieve this goal. For this purpose we quality control our results using tests for covariance stationarity, atmospheric connectivity between emissions from the surface and their measurement in the boundary layer, and the flux random error. We already describe these procedures in Sect. 2.5. Flux calculation.

We believe that Sect. 3.2 Flux footprint provides a comprehensive account of the monitored footprint to explain the observed fluxes. There, we display and discuss the campaign mean flux footprint climatologies for the winter and summer field campaigns. Because we adopted the footprint results of Squires et al. (2020), we point to their publication for even more information. We feel that reciting additional details of the Squires et al. (2020) study would be at the expense of conciseness and accessibility for a substantial fraction of the readership.

**• The writing needs some improvement.**

Response: The manuscript has been proofread and the writing has been improved.

**Specific comments (page, line)**

**2,35. The abstract can be shorter. The first paragraph provides generic information. For example, the severe air pollution problem of Beijing is well known, and there is no need to specify in which institute the measurements were conducted.**

Response: As suggested the abstract has been shortened, removing the description of severe air pollution in Beijing and the institute where measurements were made.

**2,51-52. In which aspects, magnitude or/and temporal distribution?**

Response: Magnitude. This has been clarified in the text.

**2, 68-69. Urban emissions have been widely evaluated through a number of direct and indirect methods. A brief discussion on the emissions data provided by those methods in comparison to the data obtained from eddy covariance flux towers will put in context the information presented in this study.**

Response: This line has been removed and the discussion of urban VOC emission has been expanded. See response to previous comment for details.

**3, 72-74. Be more specific, vehicular exhaust no. 1 and vehicular exhaust no. 2 do not provide major information.**

Response: The factor names as determined by Wang et al. (2015) and quoted here are: "gasoline evaporation and vehicular exhaust no. 1" and "vehicular exhaust no. 2". Both factors contain compounds associated with vehicular exhaust (pentanes, acetylene, benzene, and toluene in factor 1; acetylene and C2-C4 alkanes in factor 2). The compounds associated with each factor have been added to the manuscript.

"A positive matrix factorization (PMF) model applied by Wang et al. (2015) identified four major sources of VOCs in Beijing: two transportation factors denoted "gasoline evaporation and vehicular exhaust no. 1 (containing pentanes, acetylene, benzene, and toluene)" and "vehicular exhaust no. 2 (acetylene and C2-C4 alkanes)", "natural gas and liquid petroleum gas use and background" and "paint and solvent use and industry".

**3, 78-79. One of the main goals of the quality monitoring is to evaluate the effectiveness of the control measures in place. Many cities around the world have made important investments to build air quality monitoring networks to measure and report hourly concentrations of key airborne pollutants, including particulate matter and speciated VOCs.**

Response: It's true that the evaluation of the effectiveness of control measures is one of the main objectives of air quality monitoring. We feel that it is fair to say that it can be difficult to determine the impact of specific measures. The large scale short term measures applied in Beijing are unusual and warrant discussion in the context of this paper.

**3, 93. Activity data?**

Response: "Activity factors" changed to "Activity data"

**3, 95-97. The application of the eddy covariance method to evaluate the accuracy of gridded emission inventories of VOCs in urban areas was done by first time in Mexico City (see Velasco et al., 2005, 2009).**

Response: This sentence has been removed and reference to Velasco et al., 2005 and 2009 has been added to section 3.5.

**3, 99-106. A statement of the hypothesis to test including the study objectives in context of the APHH project will help to analyze the flux observations and reach strong conclusions.**

Response: A statement of the objectives of this study has been added to the introduction.

**"**The objectives of the study were to measure the surface-to-atmosphere fluxes of VOCs using the method of eddy covariance at a site in Beijing where a very comprehensive suite of atmospheric, physical and chemical parameters were assessed by the wider APHH-Beijing project team, and to use these data to assess the validity of a widely used emissions inventory"

**4, 100-119. Why was the AIP site selected to deploy an eddy covariance system? Does it meet the eddy covariance assumptions of homogeneity in terms of land cover and emissions distribution? A much more comprehensive description of the monitored site is needed. Information of the land cover distribution, urban morphology, roughness elements, trees characteristics, vehicular traffic, population density, etc. is needed. Please, consider the Local Climate Zone classification (see Stewart and Oke, 2012).**

Response: This section has been updated to provide a more detailed description of the site.

"Measurements of VOC mixing ratios and fluxes were made during two intensive measurement campaigns (winter: 12/11/2016 - 10/12/2016; and summer: 15/05/2017 - 24/06/2017) from a 325 m

high meteorological mast located on the campus of the Institute of Atmospheric Physics, Chinese Academy of Sciences (IAP) in Beijing (39°58'33"N 116°22'41"E). This site was used as the main sampling site for all APHH-Beijing programme activities (see Shi et al., 2019). The campus is situated north of central Beijing between the 3$^{rd}$ and 4$^{th}$ ring roads, with parkland to the east and west and a mix of dense residential and commercial (restaurants and shops) buildings to the north and south. Busy roads are situated 120 m north and 300 m east of the mast. For comparative purposes, the sampling location can therefore be described as being "urban background" with respect to ambient concentrations for central Beijing with the main sampling inlet being significantly elevated above ground level. Flux measurements of $CO_2$ have been made previously at this site (Song and Wang, 2012; Liu et al., 2012a; Song et al., 2013). The fractional land cover within 2 km of the tower was assessed by Song and Wang (2012) and found to be buildings (0.65), vegetation (0.23) and roads (0.12). Using the local climate zone (LCZ) classification system described by Stewart and Oke, (2012) this area can be classified as LCZ2 "compact mid-rise" becoming LCZ1 "compact high-rise" to the south west. The aerodynamic roughness lengths were calculated by Li et al. (2003) as cited by Liu et al. (2012) and found to be 2.5, 3.0, 5.3 and 2.8 m from the north east, south east, south west and north west respectively with the zero-plane displacement heights of 12.3, 15.0, 26.4, and 13.2 m respectively. The flux inlet was located at 102 m, this is over twice the mean height (45 m) of the tall residential building to the southwest (Liu et al., 2012a) and below the mixing layer height (Squires et al., 2020) positioning the inlet in the constant flux layer."

**4, 117-118. Urban background in terms of ambient concentrations or urban fluxes?**

Response: With respect to ambient concentrations. This has been clarified in the text.

**4, 126. '… at a height of 102 m above ground level' was mentioned in the previous paragraph.**

Response: This has been removed from the previous paragraph.

**5,155. The legend should guide readers to understand the sketch of the sampling system presented in this figure without need of going back to the text.**

Response: The legend has been expanded to give a description of the inlet setup.

"Figure 1: Schematic of the gradient sampling system. The PTR-ToF-MS inlet was switched between a common flux inlet line (40 min) and a gradient switching manifold (20 min) in an hourly cycle. The gradient system sampled from five heights into 10 stainless steel containers."

**5, 162-163. You may consider moving this table to the main article since it is the only material presented as Supplementary Information.**

Response: In response to a comment below the supplementary information has been expanded so this has been left in the SI.

**7, 200. The pioneering studies in Mexico City mentioned in a previous comment were done using the disjunct eddy covariance method and the original PTR-MS instrument.**

Response: Thank you for drawing our attention to these studies, references to Velasco et al. 2005 and Velasco et al. 2009 have been added here.

**7, 201. A detailed description of the flux data postprocessing? The eddy covariance method does not estimate fluxes, does measure fluxes.**

Response: "estimation" has been changed to "measurement". The flux data post-processing is described in the following paragraphs (lines 225 to 249 in the ACPD version). Storage below the measurement height was calculated and added to the flux. The flux limit of detection was calculated and fluxes failing the stationarity test described by Foken and Wichura (1996) and falling below a $u_*$ threshold of 0.175 m s$^{-1}$ were filtered.

**7, 203-204. Turbulent fluxes are computed using instantaneous deviations or fluctuations of the vertical wind velocity and VOC mass density (e.g., mg m-3).**

Response: This has been corrected to show that turbulent fluxes were computed using instantaneous deviations or fluctuations of the vertical wind velocity and VOC concentration (nmol m$^{-3}$).

**7, 211-212. What about the time taken by the instrument to analyze the sample?**

Response: The PTR-ToF-MS analysed the sample at 5Hz (0.2 s) which is low compared to the time taken for the sample to travel down the inlet line.

**7, 228-229. Why did Squires et al. (2020) choose this method?**

Response: The method was chosen as it could be applied to all instruments measuring fluxes during these campaigns. For some species gradient data was available which could have been used to calculate a storage flux but this would have prevented direct comparison of fluxes.

**7, 240-241. Do you mean stationarity?**

Response: Yes, this has been corrected.

**7, 242. Explain briefly the stationarity test.**

Response: A description of the stationarity test has been added.

"In order to calculate a reliable flux, the averaging period must meet stationarity requirements such that this period is shorter than the time scale at which changes in atmospheric conditions occur. The stationarity of the flux across the half hour averaging period was assessed using the method described by Foken and Wichura (1996). The half hour averaged fluxes were filtered where the stationarity criterion (the difference between average flux and mean average flux of its components) exceeded 60% and were above the limit of detection."

**7, 245-247. There are methodologies available to fill gaps in time series of turbulent fluxes measured by eddy covariance. However, depending on the study purposes it may not be necessary to fill such gaps. Consider that in urban environments VOC fluxes respond mainly to anthropogenic and not natural variables like in a forest. Human actions can be random at times (e.g., traffic jams), thus very low or very high fluxes are possible.**

Response: While gap filling methods exist for vegetation, gap filling urban data is harder. As the reviewer states, human actions can take place at unpredictable times and both high and low fluxes are possible. Most data falling below the $u_*$ threshold occurred at night when fluxes were low. The decision was taken to substitute these data with campaign average values for those hours so as not to artificially raise the campaign average data.

**7, 238-249. Please, indicate the number of periods excluded for further analysis due to lack of stationarity and/or enough turbulence. How many averaging periods were affected by rain/snow and instrument maintenance?**

Response: In the summer field campaign, 12% of the flux averaging periods fell below the $u_*$ filter. Of the masses contributing more than 0.75% to the total flux, the average percentage of flux files filtered for failing the stationarity test is 25%. This ranged from 52% for formaldehyde to 13% for methanol.

In the winter campaign, 25% of the flux averaging periods fell below the $u_*$ filter. Of the masses contributing more than 0.75% to the total flux, the average percentage of flux files filtered for failing the stationarity test is 69%. This ranged from 73% for methyl vinyl ketone and methacrolein formaldehyde to 62% for acetaldehyde.

29 h (6% of the PTR-ToF-MS operational period) were lost to instrument maintenance in the winter campaign and 133h (16% of the PTR-ToF-MS operational period) were lost to instrument maintenance in the summer. Precipitation was very low in both campaigns with light rain fall on the 20[th] of November and snow overnight into the 21[st]. In the summer campaign light rain was observed on 22nd and 30th May and on 2nd, 6th, 18th and 21st – 23rd June.

This has been added to the manuscript:

"Over the course of the winter campaign 29h (6% of the PTR-ToF-MS operational period) were lost to instrument maintenance and calibration and 133h (16% of the PTR-ToF-MS operational period) were lost to instrument maintenance and calibration in the summer."

"In the winter campaign, 25% of the flux averaging periods fell below the $u_*$ filter. Of the masses contributing more than 0.75% to the total flux the average percentage of flux files filtered for failing the stationarity test is 69%. This ranged from 73% for methyl vinyl ketone and methacrolein formaldehyde to 62% for acetaldehyde. In the summer field campaign 12% of the flux averaging periods fell below the $u_*$ filter. Of the masses contributing more than 0.75% to the total flux the average percentage of flux files filtered for failing the stationarity test is 25%. This ranged from 52% for formaldehyde to 13% for methanol."

**8, 254-255. This information was already provided.**

Response: This sentence has been removed.

**9, 263-264. This figure and the description of the dominant winds during the field observations are useless without a proper analysis of the land cover and distribution of the measured fluxes by wind sectors.**

Response: This figure (together with figure 3) is intended to provide an overview of the meteorological conditions during the two field campaigns. Discussion of the land cover and the flux footprint is provided in section 3.2.

**9, 265-266. This information was already provided.**

Response: This sentence has been removed.

**10, 279-281. Is this true at nighttime and daytime? Atmospheric stability evolves during the diurnal cycle, thus significant changes in the footprint can be expected.**

Response: This is the campaign average including both day and night time data. Significant changes to the footprint will occur over the course of the day but this is presented to give an overview of conditions across both campaigns.

**11, 292-296. This figure is incomplete. A color gradient to indicate the flux contribution is missing. No spatial scale was included neither a north indicator. The article is in English, but the map includes text in Chinese. The legend refers to a grid of cells, but no cells are included. Do these footprints correspond to nighttime or daytime?**

Response: The figure has been updated. This is the campaign average including both day and night time data. Each coloured pixel represents one grid cell.

[Figure]

Figure 4. The campaign mean flux footprint climatologies for the winter (left) and summer (right) field campaigns. The IAP meteorological tower is represented by the red dot and surrounding 100 x 100 m cells are coloured by their mean contribution to the flux. The contour lines correspond to the distances from the tower where the surface contributions to the measured fluxes cumulate to 30, 60 and 90% respectively. Map tile sets are © Google.

**11, 302-303. Define 'sawtooth cycle'. Although the paper is about fluxes, it will be good to provide a short description and a figure about their diurnal cycle.**

Response: The sawtooth cycle is described in the next sentence. These sentences have been combined to make this clearer. A diurnal profile of the total measured mixing ratio is shown in Fig. 6. Diurnal mixing ratios of individual compounds have not been added as detailed discussion of the mixing ratios is beyond the scope of this work.

"Mixing ratios of these compounds tracked the pollution events in a distinctive "sawtooth cycle" described by Jia et al. (2008) where mixing ratios increased over a period of 3 - 4 days when the wind speed was low and prevailing wind direction was southerly before dropping rapidly when the wind direction moved to the North West"

**12, 322-330. This paragraph should be part of the methodology. Do not repeat information.**

Response: We feel that providing a brief description of fluxes when moving from discussing mixing ratios to fluxes will help those readers who have more experience working with VOCs than micro-meteorology to interpret the flux data discussed in this section.

**13, 349-351. Turbulent fluxes measured by eddy covariance represent the VOC exchange within the observed footprint. Periods affected by advection were removed through the stationarity criterion, isn't it?**

Response: The stationarity filter will remove averaging periods affected by a change in meteorological conditions. The transport in the winter campaign is over a longer time period with a 3-4 day period of high VOC mixing ratios (lines 302-305 of the ACPD manuscript). Deposition of these VOCs is likely to reduce the net flux measured.

**13, 351-352. This is unclear. Do you mean that turbulent fluxes cannot be measured during winter in Beijing, and therefore half of the study was useless?**

Response: The fluxes measured in the winter campaign are valuable as they show a low net VOC flux. However, as local emission cannot be disentangled from the deposition of VOCs transported from outside of the footprint, these data cannot be used to assess local emission.

**14, 366. Table 2. A complete list of the 50 VOC species identified and evaluated in this study should be included in the Supplementary Material. You may use as reference how Karl et al. (2018) analyzed and presented an even larger list of VOC species.**

Response: As suggested a list of VOC species has been added to the supplementary information.

**14, 378-382. This section discusses fluxes no mixing ratios. Mixing ratios were discussed in section 3.3.**

Response: This paragraph has been moved to section 3.3.

**14, 380-382. London and Manchester are only two cities. There are many papers reporting mixing ratios of oxygenated VOCs for cities of China and the rest of the world.**

Response: References to studies carried out in Beijing (Shao et al., 2009) and Ahmedabad (Sahu and Saxena 2015) have been added.

**15, 387. Vehicular traffic is a major source of isoprene in urban areas (e.g., Borbon et al., 2001; Jaimes-Palomera et al., 2016).**

Response: These papers have been added to the discussion.

"Isoprene, which has both biogenic and vehicular sources (Borbon et al., 2001; Jaimes-Palomera et al., 2016), had median mixing ratios of 1.08 and 0.38 ppb in the winter and summer campaigns respectively."

**17, 400. Do you mean less than 1%? Figure 6 has not been cited yet in the text.**

Response: "VOC species which contribute more than 0.75% to the total measured VOC flux" is correct. Figure 6 is cited on line 375 of the ACPD manuscript.

**17, 403. Be more specific on the type of combustion sources (e.g., vehicular traffic, industry, biomass burning).**

Response: Changed to read: "NOx in urban areas is a combustion product so the NOx flux is used here as proxy for vehicular emissions"

**17, 409. A vehicular emission control program?**

Response: Changed to read "Beijing operates a motor vehicle emissions control program…"

**17, 412-415. A comparison against the fluxes reported by Velasco et al. (2005, 2009) for two locations of Mexico City will help to put in context the fluxes observed in Beijing with respect to other large city of a developing nation.**

Response: A comparison with the fluxes recorded in Mexico City has been added.

"The toluene flux is, however, significantly lower than the 2.5 and 9.2 nmol m$^{-2}$ s$^{-1}$ recorded at two sites in Mexico City (Velasco et al. 2005; 2009). Velasco et al. (2009) identified both exhaust emissions and evaporative sources contributing to the toluene flux. The addition of these evaporative sources in Mexico City as well as differences in vehicle fleets is likely to explain why these fluxes were larger than those recorded in this study and in the UK."

**17, 416-418. How do these ratios between mixing ratios and fluxes compare to those reported in the literature for other cities?**

Response: The discussion of the B/T ratio has been expanded, this is discussed in more detail by Squires et al. (2020)

"Squires et al. (2020) also concluded that the benzene/toluene ratio, which for concentrations was 0.89 in the winter and 0.73 in the summer was within the range expected for primary exhaust emissions but higher than the 0.6 calculated by Barletta et al. (2005) for Beijing. For fluxes, the benzene/toluene ratio was 0.72 in winter and 0.31 in summer (Squires et al. 2020). The summer value is lower than expected for primary exhaust emissions, suggesting that fuel evaporative loss contributes to aromatic VOC emission in summer (Squires et al. 2020)."

**17, 420-425. Propene is dominated by emissions from vehicular traffic in locations with heavy traffic (see Velasco et al, 2005, 2009). Averaging periods affected by large plumes are expected to be removed by the stationarity criterion. If stacks of large industries were within the monitored footprint and their impact was not filtered by the stationarity criterion, the turbulent fluxes reported in this study are suspicious.**

Response: The correlation with NO$_x$ suggests that at the IAP site emission was dominated by traffic. The reference to industrial sites was made to provide context. The sentence has been updated to make it clear we are not proposing emission from heavy industry at this site. Reference to has been made to Velasco et al (2005, 2009).

"The fluxes of other compounds shown to correlate with the flux of NO$_x$ were propene and methanol with R$^2$ values of 0.63 and 0.71. Propene is known to be emitted by vehicular traffic (Velasco et al. 2005; 2009) and the correlation with NO$_x$ suggests that traffic is the main source at this site. Propene has also been recorded in plumes of VOCs released from industrial activity (Karl et al., 2003), but no large industrial sites were present within the flux footprint of the IAP tower."

**17, 430-432. See Fig. 4 in Velasco et al. (2005) and Fig. 3 in Velasco et al. (2009).**

Response: Comparison with these studies has been added.

"These methanol emissions are comparable to those observed in London and Mexico City with 2.5 nmol m$^{-2}$ s$^{-1}$ (August to December) recorded by Valach et al. (2015), 8.2 nmol m$^{-2}$ s$^{-1}$ (October) reported by Langford et al. (2010) in London, and 9.1 and 12.8 nmol m$^{-2}$ s$^{-1}$ (April and March respectively) reported in Mexico City (Velasco et al., 2005; 2009)."

**19, 445. The acronyms POCP and PPCP have not been defined in the text yet.**

Response: These acronyms have been explained in the caption: "Figure 6. Total observed VOC concentration (a) and flux (b) measured during the summer field campaign, scaled using photochemical ozone creation potentials (POCPs) (c, d) and photochemical PAN creation potentials (PPCPs) (e, f)."

**20, 446-449. A similar figure showing the flux correlations between species during the winter period is needed.**

Response: The VOC fluxes recorded in the winter campaign were low with a lot of noise. Therefore a correlation plot would not be especially meaningful. We took the decision to focus on the summer campaign where clear VOC fluxes were observed (line 351).

**20, 466-467. See previous comment on the vehicular contribution to urban isoprene.**

Response: The following text has been added:

"Globally, plants are the main source of VOCs into the atmosphere, with isoprene and monoterpenes the main VOC species emitted (Guenther et al., 1995; 2012). However, in urban areas, isoprene is present from both biogenic and vehicular sources (e.g. Borbon et al., 2001; Langford et al., 2010; Valach et al., 2015; Jaimes-Palomera et al., 2016)."

**21, 481-482. How and who determined this emission factor for Beijing's urban vegetation?**

Response: This emission factor is the default value used in MEGAN. The emission factors used by MEGAN were calculated by combining isoprene observations from numerous enclosure studies with landcover data from ground measurement inventories, satellite based inventories, and ecoregion descriptions. Calculation of emission factors is described by Guenther et al. (2006).

**22, 511-514. Based on the comprehensive set of measurements conducted during the APHH study, is it possible to adjust the POCP and PPCP factors for Beijing's conditions? If it is not possible, wouldn't be better to evaluate only the OH reactivity of each compound?**

Response: In response to a comment by reviewer 2, the description has been expanded to include discussion of the conditions used by Derwent et al. (1998). The conditions observed in Beijing were within the range of NOx values tested by Derwent et al.

"Derwent et al. (1998) used NOx levels of 0.014 mg m$^{-3}$ as the base case when determining POCPs and tested the POCP calculation at half and double these NOx levels. The median NOx concentration observed in the Beijing summer campaign was 0.0213 mg m$^{-3}$ (Squires et al., 2020), within the twice base case range."

**22, 514-516. Light alkanes can be important players if their emissions are high like in Mexico City (see Jaimes-Palomera et al., 2016).**

Response: This has been acknowledged in the text.

"Due to their low POCP and PPCP relative to alkenes and many oxygenated compounds, alkanes are likely to have a limited impact on ozone and PAN formation. However, at large concentrations they can still have a significant impact (Jaimes-Palomera et al., 2016)."

**24, 540. The VOC classification used here is 'strange'. Be more specific on the species included in the group 'other hydrocarbons'.**

Response: Groups have been reclassified as: methanol + ethanol, oxygenated VOCs, aromatics, predominantly biogenic compounds, alkenes and other chemical groups.

**24, 543. Consider a different title for this section. Why did the evaluation of the accuracy of the gridded emissions inventory was limited to the summer period?**

Response: Title changed to "Emission factor analysis and validation". The evaluation was limited to the summer period by the low net flux recorded in the winter, making it difficult to identify local emissions.

**24, 544-546. This was already mentioned in the introduction (lines 90-94). Indicate the emissions inventory's year reference.**

Response: We feel it aids the reader to give a brief description of the inventory at the start of this section. The emission inventory year reference (2013) is given on line 566 of the ACPD manuscript.

**25, 553-554. Were any power plant or large industry within the monitored footprint? If the answer is positive, the fluxes reported here might not be valid. Review the basic assumptions for measuring turbulent fluxes by eddy covariance (Velasco and Roth, 2010).**

Response: There were no power plants or heavy industry with in the footprint. The footprint was made up of parkland residential and commercial buildings with large two large roads: the Jingzang Expressway and the Beitucheng West Road. This is described in section 3.2.

**25, 575-576. The poor agreement between measured fluxes and estimated emissions might respond to the size of the grids in the emissions inventory. Cells of 9 km2 might be too large to capture the emissions observed by the flux tower.**

Response: Squires et al. (2020) tested the sensitivity to location by shifting the grid 3 km in each direction, finding the same overestimation of $NO_x$ and CO for all directions. This suggest that there is an issue with the proxies used to scale emission to the city centre rather than the grid cells being too large to capture emissions at the IAP site.

**27, 590. According to the industrial emissions reported in the inventory developed for air quality modelling and presented in this figure, the site selected to conduct flux measurements by eddy covariance in Beijing is not adequate. Review the conditions needed to perform representative eddy covariance flux measurements in urban areas (Velasco and Roth, 2010).**

Response: While the inventory assigns large industrial emissions to this area, no large factories or industrial sites were present, making it a suitable site for eddy covariance measurements. In response to a previous comment the site description has been expanded to give a more thorough description to the fractional land cover. The large industrial emissions present in the inventory are likely to be caused by downscaling the inventory using proxies such as population. This misallocation of emissions in the inventory is discussed in lines 583-586.

**28, 597-599. If it was known in advance that the inventory did not include the biogenic component, then which was the purpose of comparing the measured fluxes of isoprene? The emissions estimated by Megan should be added to those reported by the MEIC inventory.**

Response: Comparison with isoprene is made for the sake of completion. As suggested the emission predicted by MEGAN has been added to this figure and the text updated.

"Isoprene emissions are significantly underrepresented in the inventory with measured fluxes 20 times higher than inventory emissions. This is expected as the inventory only considers anthropogenic emission and in the summer isoprene emissions are dominated by predominantly-biogenic isoprene. Adding the isoprene emission predicted by MEGAN to the MEIC emission (Fig. 10h) shows that the

inclusion of biogenic isoprene enables a more accurate estimate of total isoprene emission. The park at the base of the tower means that a larger biogenic source may be present at the IAP site than would on average be expected for Beijing."

[Figure]

"Figure 10. Comparison of 9 VOC classes from the MEIC emissions inventory (Table 3) with measured VOC flux (black line). The diurnal profiles of the isoprene emission predicted by MEGAN using the optimised emission factor is shown in grey (h)."

**28, 608-610. Indeed, the presence of methanol, aldehydes and organic acids can be explained by local atmospheric chemistry, however, it was previously proved that the time taken by an air parcel to reach the top of the tower was much less than the time needed for the oxidation of such compounds. If it was not the case, fluxes of those compounds cannot be measured by eddy covariance in Beijing.**

Response: We thank the reviewer for pointing this out. The text has been amended to reflect the fact that formation in the atmosphere will not contribute significantly to the flux.

"It is possible that some underestimation of methanol, aldehydes and organic acid fluxes can be explained by their formation on surfaces following the oxidation of primary-emitted predominantly-biogenic (e.g. Lee et al. 2006; Acton et al., 2018) or anthropogenic compounds (Atkinson, 2000). However, given the ~68 s time taken for an air parcel to reach the inlet point, it is not likely that formation in the atmosphere contributes significantly to the flux of these compounds. Consumer goods are known to be a large source of oxygenated VOCs (Dinh et al., 2015; Nematollahi et al., 2019). The use of these products is likely to vary considerably between households making bottom-up emission hard to estimate."

**29, 617. Instead of presenting conclusions and final remarks, this section is a summary of results.**

Response: This section has been retitled "Summary"

**29, 618. Rewrite this sentence. For example: 'Fluxes of speciated VOCs at district scale were measured by first time in Beijing ….'.**

Response: Re-written as suggested.

"Fluxes of speciated VOCs at district scale were measured for the first time in Beijing with emissions dominated by small oxygenated compounds methanol, acetaldehyde and ethanol and formic acid which could not be separated"

**29, 625. 'Stable atmospheric conditions' may sound better.**

Response: Changed to "Stable atmospheric conditions"

**29, 626-627. It may be true for VOC species emitted by evaporation of fossil fuels and solvents, but not for species emitted by other emission sources. It is not possible to talk about advection of VOCs in the case of fluxes measured at district level by eddy covariance, see previous comments.**

Response: As stated in response to the previous comment this does not refer to advection within the flux averaging period but transport over a period of days leading to a high concentration of VOCs in the atmosphere dampening the net flux due to local emission.

**29, 630-633. See previous comments about comparing the fluxes reported in this study with those observed in Mexico City.**

Reference: Reference to this study has been added: "Both fluxes and mixing ratios of toluene were lower than those recorded in Mexico City (Velasco et al., 2009)".

**Response to RC2: Anonymous Referee #2**

**This manuscript presents the VOC concentration and flux measurements using a PTRMS at a tower in Beijing during the China-UK APHH summer and winter campaigns. Large concentrations and fluxes of small OVOCs were measured in both winter and summer. The measurements also show high concentrations of isoprene and aromatics in both summer and winter, although the surface-to-air fluxes were small. The authors compared the results to the fluxes predicted by the MEIC (anthropogenic) and MEGAN (biogenic) emission inventories and attempted to explain some of the discrepancies. The topic of the manuscript is of interest to the community and certainly within the scope of ACP. The measurement methodology and analytical methods are standard. Most of the results appear reasonable. However, the interpretation of the isoprene and aromatic sources have some contradictions with previous studies and the known characteristics of the urban site, warranting more in-depth analysis. Also, the writing has a fairly large number of errors in grammar and punctuation usage. The senstences were often too long and commas are often missing. I therefore recommend that the paper undergo major revision to address some of the key issues and also to improve the writing.**

Response: The authors thank the reviewer for taking the time to review this manuscript and for their constructive comments. The interpretation of isoprene and aromatic compounds has been expanded (see responses to specific comments below) and, as well as addressing the sentences identified by the reviewer, the manuscript has been proofread to improve the writing.

**Major comments:**

**Figure 3a: Why was the timeseries of the meterological variables drawn for November 16 to December 7 for the winter campaign? This does not match the duration of the winter campaign nore that of the PTR-MS measurements stated in lines 254-255.**

Response: The campaign dates (line 254) in the APCD manuscript are those of the broader APHH-Beijing winter campaign (Shi et al. 2019). The meteorological variables plotted in Figure 3a began when the sonic anemometer used during this campaign was set up. The start of the PTR-ToF-MS operational period was delayed by technical issues with the instrument setup. PTR-ToF-MS measurements therefore started later, as stated on line 255.

**Lines 308-309: "In addition, isoprene, which has both biogenic and anthropogenic sources, had a median mixing ratios of 1.08 and 0.38 ppb in the winter and summer campaigns respectively.": This is an interesting result. Although several studies have indicated possible anthropogenic emissions of isoprene (potentially related to traffic), the origin of this anthropogenic isoprene have not been fully explored. Most researchers still predominantly associate isoprene with biogenic emissions. Also, I do not think I have ever seen isoprene concentrations this much higher in winter (median 1.08) than in summer (median 0.38). Could the authors elaborate a little on this anthropogenic isoprene? Were there strong correlations between isoprene concentrations or fluxes with other source-specific VOCs? Figure 5 shows that there was not a strong positive flux of isoprene in winter. This contradicts (1) the findings from previous studies suggesting that the anthropogenic isoprene is traffic-related, and (2) the fact that the IAP tower is located near major roads with heavy traffic (major ring roads within its 1-2 km flux footprint). Also, it would be helpful to cite some key reference on the anthropogenic isoprene source in lines 308-309.**

Response: The discussion of isoprene mixing ratios has been expanded with the following text added:

"High winter isoprene concentrations have been observed previously in Beijing (Li et al., 2019) where an average value of 1.01 ppb was observed as well as during haze periods (Sheng et al., 2018). High winter isoprene has also been reported in other Asian countries with isoprene mixing ratios of 1.6 and 1.1 ppb recorded in Ahmedabad (Sahu and Saxena 2015) and Kathmandu (Sarkar et al., 2016). These studies are all characterised by the use of a PTR-MS, it is possible that an isomer of isoprene contributes to the high winter time concentrations observed in Asian cities. In contrast, other studies have reported higher summer time isoprene concentrations in Beijing (e.g. Cheng et al., 2018). In the winter, isoprene mixing ratios peaked during haze events and correlated well with anthropogenic and oxygenated VOCs. $R^2$ values of 0.82, 0.77 and 0.74 were observed for the correlation of isoprene with propenal, benzene and toluene respectively."

The weak isoprene flux in the winter does not necessarily contradict the high mixing ratios observed in the winter. As isoprene correlates well with the haze periods and other VOCs this suggested that high concentrations are a result of transport from outside the flux footprint and not enhanced local emission. This is discussed on lines 465-468 of the ACPD paper:

"In the winter campaign, emissions of both isoprene and monoterpenes were comparatively small with average fluxes of 0.79 and 0.06 nmol $m^{-2}$ $s^{-1}$, respectively. Despite the weak isoprene flux, mixing ratios were larger in winter than in summer with mixing ratios consistent through the day at an average value of 1.21 ppbv. The high isoprene mixing ratio in the winter suggests transport of anthropogenic isoprene from outside of the flux footprint together with a lower rate of photochemical loss than in the summer."

**Lines 346-348: "The weak VOC flux observed in the winter may, in part, be caused by low volatilisation of VOCs in winter conditions, where the average temperature was 3.6 ◦C compared to 25 ◦C in the summer.": But most of the the species shown in Fig 5 and Table 2 should be sufficiently volatile even at 3.6oC (e.g., methanol, ethanol, acetaldehyde, etc). I think, to be more illuminating, the authors should state the known major sources of these low-molecular-weight organics and indicate whether those sources are smaller in winter.**

Response: The winter campaign is characterised by large pollution events controlled by meteorological processes (lines 301 to 305 of the ACPD manuscript). As we are recording the net flux of VOCs it is likely that deposition of VOCs transported from outside the flux footprint will dampen the emission flux from local sources recorded (lines 348-349 of the ACPD manuscript). While small OVOCs will be sufficiently volatile even at low temperatures, the low temperatures observed in the winter campaign will reduce the rate of evaporation. As suggested, a short description of possible sources has been added. This section has been reordered to put more focus on the impact of transport from outside the footprint.

"In the winter, the high concentrations of VOCs at this site, mainly resulting from advection from outside the city, might lead to deposition of VOCs. This would have the effect of suppressing the net VOC emission rate measured at the site. The average temperature in the winter campaign was 3.6 °C compared to 25 °C in the summer, and this would result in lower volatilisation of liquid VOCs (for example in gasoline), supressing the emission rates. Sources such as cooking and solvents in cleaning products, cosmetics and paints contribute to urban VOC emission (Karl et al., 2018) and these sources are likely to be reduced in the winter when cold weather causes ventilation of buildings to be reduced. Measured VOC emissions from indoor and outdoor sources may therefore be lower in

winter. In addition the lower atmospheric turbulence in winter and generally lower fluxes resulting in greater uncertainty in our winter fluxes compared to summer fluxes."

**Lines 380-384: What seasons were these measurements conducted?**

Response: Updated to read "These compounds have been observed at high concentrations in many cities (Langford et al., 2009 (summer); Valach et al. 2015 (August-September)) with ethanol reported to be the most abundant VOC in London with a mean mixing ratio of 5 ppb in summer and winter (Dunmore et al., 2016)"

**Lines 386-388: If the anthropogenic contribution of isoprene is so significant at this site in winter (especially given the much larger concentration in winter than in summer), how can one catagorically label isoprene as biogenic, even in summer?**

Response: This is correct and has been reworded as "predominantly biogenic". It should be noted that there is a difference between the concentration, which in the winter is mostly a result of regional transport, and the flux, which is emitted locally. The large winter concentration does not imply a large local emission. Given that the summer isoprene flux shows a clear diurnal profile which fits a biogenic model (Fig. 8) and no clear rush hour peaks we feel it can be labelled as predominantly biogenic.

**Lines 399-401: "The relationship between the fluxes of VOC species which contribute more than 0.75% to the total measured VOC flux and 400 the NOx flux in the summer measurement period is shown in Fig. 7.": This sentence is grammatically incorrect and therefore impossible to understand. Please kindly revise.**

Response: Re-worded to read: "The correlation between (a) the NOx flux and (b) those VOC species that contribute more than 0.75% to the total measured VOC flux is shown in Fig. 7."

**Lines 402-403: "NOx in urban areas is a combustion product so the NOx flux is used here as proxy for combustion sources of pollutants": At this site, the NOx flux is most strongly a proxy for vehicular emissions.**

Response: "NOx in urban areas is a combustion product so the NOx flux is used here as proxy for combustion sources of pollutants" changed to read "In urban areas NOx is also exclusively produced by combustion sources of which motor vehicles are the main source. The NOx flux is therefore used here as a proxy or vehicular emissions."

**Lines 426-428: "The close correlation with the NOx flux and aromatics such as toluene suggests a combustion source contributes to the total methanol flux but the summer methanol flux also correlates well with small oxygenated VOCs such as acetaldehyde.": What is the evidence for the first half of this statement? Particularly since it contradicts the second half of the sentence.**

Response: The correlation with $NO_x$ is shown in Fig. 7. There is not a close correlation with toluene, we thank the review for pointing out this error. This has been rewritten:

"The close correlation with the $NO_x$ flux suggests a combustion source contributes to the total methanol flux but the summer methanol flux also correlates well with small oxygenated VOCs such as acetaldehyde (Fig. 7)."

**Section 3.4.3: Again, I think the source of isoprene at this IAP site should be more carefully interpreted, given that a large concentration was observed in winter.**

Response: A more detailed discussion of the large winter concentration has been added to section 3.3 (see response to second comment).

**Lines 468-470: "The low winter isoprene flux suggested a small contribution of anthropogenic isoprene to the total flux.": I am not convinced. Previous studies indicate that the anthropogenic emission of isoprene was vehicular. And this IAP tower site should be strongly affected by traffic emissions.**

Response: Anthropogenic isoprene is known to be emitted from traffic (Borbon et al., 2001). This has been added to the discussion. The correlation of VOC fluxes recorded in the summer campaign is shown in Fig. 7. Isoprene shows a very poor correlation with traffic tracers such as toluene, C2-benzenes ($R^2$ 0.26 and 0.25) and $NO_x$ ($R^2$ 0.38) which suggests that the contribution from traffic emissions to the summer isoprene flux is low compared to the biogenic source. In addition to this the isoprene flux profile could be modelled well using MEGAN (a biogenic isoprene emission model), see Fig. 8. We therefore feel that it is sensible to assume that biogenic emission is the dominant source of isoprene in the summer campaign. A discussion of the isoprene correlation with aromatics and $NO_x$ has been added to the manuscript:

"The summer isoprene flux correlated well ($R^2$ 0.85) with the flux of the isoprene oxidation products methyl vinyl ketone and methacrolein (MVK+MACR) and poorly with traffic tracers such as toluene, $C_2$-benzenes ($R^2$ 0.26 and 0.25) and $NO_x$ ($R^2$ 0.38) (Fig. 7). This suggests that the contribution from traffic emissions to the summer isoprene flux is low compared to the biogenic source."

**Lines 485-488: "The flux predicted by MEGAN over-estimated the observed flux with the modelled mean diurnal isoprene flux peaking at 21 nmol m-2 s-1 compared to the measured isoprene flux which peaked at 14 nmol m-2 s-1 (Fig. 8b).": How does the measured flux in summer compare with the top-down estimates using satellite-based observations by Cao et al. (2018)?**

Response: Cao et al. (2018) report an annual isoprene emission for China of 9.6 Tg $yr^{-1}$, this equates to 0.47 nmol $m^{-2}$ $s^{-1}$. The median summer isoprene emission reported in this study (Table 2 ACPD) is 2 nmol $m^{-2}$ $s^{-1}$. Given that the Cao et al. (2018) value is an annual average across a large country with many different ecosystems, these values appear broadly sensible.

**Lines 512-514: "The POCPs and PPCPs of 120 organic compounds were reported by Derwent et al. (1998) for UK conditions of the 1990s, and in the absence of more Beijing-specific data these were used to scale both the mixing ratios and the fluxes emitted within the flux footprint of the individual observed VOCs." What were the NOx levels used in the Derwent et al. (1998) calculation, and how different was it with the present-day Beijing NOx levels? Also, how long were the POCPs and PPCPs calculated for in Derwent et al. (1998), i.e, were these ozone produced over a day or several days? This makes a difference because methanol and other VOCs measured in abundance in this campaign have lifetimes > 1 day. Using 1-day POCPs and PPCPs would estimate the ozone and PAN production potentials on a local scale.**

Response: Derwent et al. (1998) used NOx concentration of 0.014 mg $m^{-3}$ as the base case but tested POCP calculation at half and double these $NO_x$ levels, reporting that "The methodology used to generate the POCPs therefore appeared to return self-consistent values over a factor of four variation in $NO_x$ emissions which confirms their robustness over a wide range of European conditions." The median $NO_x$ concentration observed in the Beijing summer campaign was 0.0213 mg $m^{-3}$, within the twice base case range tested by Derwent et al. (1998). The model runs used by Derwent et al. (1998) were 5 days long. The following description has been added to section 3.4.4:

"Derwent et al. (1998) used NOx concentrations of 0.014 mg m$^{-3}$ as the base case when determining POCPs and tested the POCP calculation at half and double these NOx levels. The median NOx concentration observed in the Beijing summer campaign was 0.0213 mg m$^{-3}$ (Squires et al., 2020), within Derwent et al.'s (1998) twice base case range."

**Lines 547-549: "Whilst activity data is often relatively well constrained at national level, its spatial disaggregation often relying on simplified proxies such as population density, adds significant additional uncertainty for the estimate of the emission for a given location.": The emission ratios (i.e, source species profile) for different NMVOC species may present an even larger uncertainty to the final inventory.**

Response: This has been acknowledged in the text: "The emission ratio of different VOC species present within each grouping could represent a source of error in the inventory."

**Lines 575: "Measured emissions of aromatic compounds were 3% and 4% of those predicted by the inventory for TOL and XYL (low mass and high mass aromatics).": This is very surprising, unless the high-resolution version of MEIC placed a disproportionally large emission near the IAP site. Several previous studies have indicated at least some underestimation of Chinese aromatic emissions in MEIC or its predecessor inventory (Liu et al., 2012; Cao et al. 2018).**

Response: We agree with the reviewer that the high-resolution version of MEIC is placing a disproportionally large emission of aromatics near the IAP site. A similar overestimation was reported by Squires et al. (2020) for NO$_x$ and CO. This is in part caused by a large industrial source MEIC predicts for the IAP, which seem unrealistic given the residential and commercial building surrounding the site. This is described in lines 575 to 589 in the ACPD manuscript. Reference to the underestimation has been added to the discussion:

"Previous studies have shown that the inventory underestimates aromatic emission at a China scale (Liu et al., 2012b; Cao et al., 2018), again indicating that the overestimation observed here is a result of the downscaling of the inventory."

**Lines 579-580: "Industrial emissions make the largest contribution to aromatic VOC emission in the inventory.": Is this statement drawn from the numbers for the local grids, or from the national inventory? The authors measurements clearly show that the aromatics were traffic-related, and I think the same was true in the MEIC inventory.**

Response: This was drawn from the local grid cells with the contribution of each grid cell weighted by the flux footprint in that cell. As shown in figure 10 the largest inventory contribution to "TOL" and "XYL" is industrial. We agree with the reviewer that the main local source of these compounds will be traffic but this is not currently represented in the inventory.

**Section 3.5: There has been several studies using formaldehyde and glyoxal observed by satellite instruments to constrain NMVOC emissions in a top-down fashion. While the resolutions were very different from those here, it might still be helpful to compare their results with the measurements here.**

Response: Reference to satellite studies has been added when discussing aromatics (see response to previous comments). As the resolution of the satellite studies differ greatly from the flux footprint, we feel direct comparison of individual compounds will not add significantly to the interpretation of individual fluxes.

**Lines 646-649: "The relatively small emissions of anthropogenic VOC species from central Beijing compared to the large mixing ratios observed suggest that the scope for policy interventions focusing on VOC emission from central Beijing is limited and that the focus must therefore be on emissions controls in regions surrounding the megacity.": Is this conclusion consistent with the observed high toluene emission and its high correlation with NOx (traffic)?**

Response: While toluene emission is significant, it is comparable to that observed in London (Valach et al., 2015) and lower than that observed in Mexico City (Velasco et al., 2009) so is therefore not large in comparison to other major urban areas. The winter concentration, while lower than Mexico City, is high compared with European cities, suggesting that a significant contribution to the total toluene concentration is coming from outside of the city. This is supported by the fact that in winter concentrations of aromatic compounds follow the "saw tooth" pattern (line 303) common to many pollutants in Beijing. These pollution events are thought to be meteorologically controlled and represent transport from outside the city instead of local emission (this is discussed in more detail by Squires et al., 2020).

**Minor comments:**

**Line 129-130: Is it possible that there is a negative bias of semi-volatile organics by filtering PM2.5?**

Response: Most semi-volatile compounds likely to be lost to filtering PM2.5 would be too heavily oxygenated to be detected by the PTR-ToF-MS 2000 used in this study. This could be an issue if using an instrument more sensitive to higher mass compounds.

**Line 233: "Loss of high frequency flux was due to measuring at 5 Hz was estimated to be less than 10%.": missing 'and' between '5 Hz' and 'was'.**

Response: First "was" deleted.

**Line 234: "Squires et al. (2020) estimated that the average the time taken for an air parcel ...": remove 'the' before 'time'**

Response: "the" removed

**Line 238: "quality-assessed"**

Response: "quality assessed" changed to "quality-assessed".

**Line 242-244: I do not understand why the quality control for flux was done on a 'file' basis. How long is the record in each file? Shouldn't the quality control be done on a certain timescale?**

Response: We agree this description is unclear. The flux files each contained half an hour of data. "Flux files" has been removed and replaced by "The half hour averaged fluxes".

**Lines 348-349: "In the winter the positive (emission) flux of VOCs from the city is likely suppressed by deposition of VOCs transported at, high concentrations, from outside of the city.": This sentence is unclear; maybe because the first comma is misplaced?**

Response: This sentence has been reworded. Together with the preceding sentence it now reads: The weak VOC flux observed in the winter may, in part, be caused by low volatilisation of VOCs in winter conditions, where the average temperature was 3.6 °C compared to 25 °C in the summer. In addition the positive (emission) flux of VOCs from the city is likely suppressed by deposition of VOCs transported, at high concentrations, from outside of the city.

**Lines 349-351: "While most mean winter VOC fluxes remained positive this term will be a balance between emission from the city and deposition of VOCs transported into the footprint region.": I do not understand this sentence. Please kindly clarify.**

Response: This sentence has been removed.

**Line 372: Missing comma after "In the summer campaign"**

Response: Comma added.

**LIne 377: "can't" should be cannot**

Response: "Can't" changed to "cannot"

[revised manuscript text omitted]

---

## Referee Report (RR1)

**Comments on the revised manuscript 'Surface-atmosphere fluxes of volatile organic compounds in Beijing' by Acton et al.**

**Reviewer: Erik Velasco**

This reviewer appreciates the corrections and additions made to this second draft. The reading improved, and many of the questions pointed out in the original revision were properly answered. However, there are still four major concerns that need to be addressed before the manuscript can be considered for publication:

- A thorough analysis of the observed fluxes needs a complete description of the monitored footprint. The current description is still incomplete. An updated description of the neighborhood characteristics is needed. The information on the aerodynamic characteristics and land cover corresponds to assessments of 10-20 years ago. Are those assessments still valid considering the accelerated changes in Beijing's urban morphology in recent years?

  Similarly, more information on the economic activities, vehicular traffic volume and composition, population density, number of trees and characteristics, etc. is needed to put in context the observed fluxes. This information will help to identify the error sources in the emission estimates of the gridded inventory.

- The manuscript reports fluxes collected during two field campaigns, a 28-day campaign in winter and a 40-day campaign in summer. However, after removing the averaging periods affected by lack of turbulence and instruments maintenance, as well as periods not meeting the stationarity criteria, only 25% and 70% of the data, respectively, were used for further analysis. This means that results are based on 7 and 27 net days of flux measurements. For the summer case, an almost one month of flux data might be enough for an initial assessment, but for the winter case, 7 days of data (for some VOC species even less) are not sufficient.

  The authors may consider the relevance of including results of the winter campaign in this article. In an urban set up, like Beijing, seven days of flux data may represent nothing else than noise. In addition, many of the analysis performed in this study were limited to the flux data collected during the summer campaign.

- The impact of regional sources of VOCs in the observed fluxes deserves further analysis and discussion. The authors need to investigate under which synoptic conditions the loss of VOCs (i.e., negative fluxes) was observed. The use of back trajectories may help to this assessment.

  Did the APHH project include measurements or modeling of the hydroxyl radical? Information on the levels of OH, NO and $O_3$ would make possible to estimate the residence time of each individual VOC species, and would support the proposed loss of VOCs. Also, it would help to evaluate the capacity of the eddy covariance system to measure fluxes of highly reactive VOCs. As indicated in the original revision, the time taken by an air parcel to reach the top of the tower could be higher than the time needed for the oxidation of such compounds, and thus the measured fluxes would not faithfully represent the emissions from the urban surface.

  The OH concentration of $1 \times 10^6$ molecules $cm^{-3}$ used to estimate the oxidation time of the assessed species is too low for a highly polluted atmosphere (see Shirley et al., 2006, for the case of Mexico City, www.atmos-chem-phys.net/6/2753/2006). A higher concentration would reduce

the lifetime of the VOCs, and therefore, the eddy covariance flux system mounted at 102 m of height might not be able of measuring fluxes of reactive VOCs.

- A final section providing conclusions and final remarks is missing. Scientific articles are expected to close with a short answer to the hypothesis to test. For example, is the eddy covariance a suitable method to measure urban fluxes of VOCs in Beijing? does the emission inventory is accurate for that particular sector of the city? What is needed to improve the inventory's accuracy?

**Specific comments**

**39-40.** Eddy covariance flux towers provide invaluable insight to evaluate the accuracy of gridded emission inventories, but they do not validate emission inventories.

**40-49.** The abstract does indicate flux measurements conducted on summer and winter, but results from the latter season are not included.

**58.** '… particulate matter of size smaller than 2.5 µm …'.

**62-63.** This statement needs a reference.

**78.** I would say 'challenging' instead of 'difficult'.

**96.** 'Cityscape scale' is a confusing term. Eddy covariance flux towers measure fluxes at neighborhood or district scale.

**97.** PTR-TOF has not been defined in the text. Define acronyms the first time they are used in the manuscript.

**98.** Eddy covariance flux towers measures fluxes, do not estimate fluxes.

**110.** See previous comment about the use of flux towers to evaluate the accuracy of emission inventories

**120.** The description of the monitored sector of the city is still incomplete. More information on the economic activities, vehicular traffic volume and composition, population density, number of trees and characteristics, distribution and variability of building heights, etc., is needed to put in context the observed fluxes, and then be able to improve the emission estimates of the gridded emissions inventory.

**121-123.** This sentence is confusing. A 325-m tall tower was used as measuring platform, but the eddy covariance flux system was mounted at a height of 102 m.

**124.** Include the height above sea level of the location where the tower was located.

**158.** Explain the purpose of the system to evaluate the vertical gradient of VOC concentrations. I may be wrong, but the text does not indicate it had been used for the flux data analysis or discussion.

**289.** Do you mean that flux data were flagged if $u* < 0.175$ m s$^{-1}$?

**340-345.** Footprints at day and night are quite different. For a thorough assessment of the hourly emissions reported in the gridded emissions inventory is mandatory to evaluate the footprint across the diurnal cycle. Do not ask readers search for such information in another article. The background map provides little information.  I cannot identify which sectors correspond to residential estates,

and which to commercial and institutional buildings, for instance. Also, I cannot identify which are primary and secondary roads. Can you mark the 3rd and 4th rings surrounding the location?

**Table 1.** Plots showing the diurnal variability of these VOC species would provide better insight in the context of the study. It does not matter, if a more detailed analysis is presented somewhere else.

**422-425.** This point deserves further discussion. I would say '… loss of VOCs by chemical reactions in the presence of fresh emitted NO ...', instead of '… deposition of VOCs …'. Did the APHH project include measurements/modeling estimates of OH? Knowing the levels of OH, NO, $O_3$, etc. would make possible to estimate the residence time of each individual VOC species.

**426-428.** I do not understand this statement. Do you mean that people cook and clean less during the winter?

**443.** It would be useful to include in the text the variability observed in the fluxes (e.g. ±1 standard deviation) for comparisons to fluxes reported in the literature.

**Figure 5**. I cannot observe any dashed line, and hardly observed the dotted ones.

**637.** More information about how the emission inventory was built is needed. It will help to understand the inventory's overestimations. Does the inventory estimate emissions based on activity data at city scale or use data for each individual grid? Which is the source of the emission factors used to estimate traffic emissions? Were the emission factors locally derived? How were the emission profiles determined? How were the evaporative emission estimated? Are there emission estimates for summer and winter?

**338.** No need to repeat the website hosting the emissions inventory. It was done in the introduction.

**667-671.** Check these two sentences. An overestimation of 3-4% is not equal to overestimate by a factor of 3-4.

---

## Author Response (AR2)

**Authors' response to: Comments on the revised manuscript 'Surface-atmosphere fluxes of volatile organic compounds in Beijing' by Acton et al.**

**Reviewer: Erik Velasco**
**This reviewer appreciates the corrections and additions made to this second draft. The reading improved, and many of the questions pointed out in the original revision were properly answered. However, there are still four major concerns that need to be addressed before the manuscript can be considered for publication:**

Response: We thank the reviewer for taking the time to provide additional comments on this manuscript and set out responses to their comments below.

- **A thorough analysis of the observed fluxes needs a complete description of the monitored footprint. The current description is still incomplete. An updated description of the neighborhood characteristics is needed. The information on the aerodynamic characteristics and land cover corresponds to assessments of 10-20 years ago. Are those assessments still valid considering the accelerated changes in Beijing's urban morphology in recent years?**

  **Similarly, more information on the economic activities, vehicular traffic volume and composition, population density, number of trees and characteristics, etc. is needed to put in context the observed fluxes. This information will help to identify the error sources in the emission estimates of the gridded inventory.**

  Response: The description of the site given by Liu et al. (2012) was consistent with the neighbourhood we observed during the fields campaign. This is supported by Cheng et al. (2018) who stated that "By the beginning of the current century, the overall height of the buildings around the meteorological tower had a fixed value, and the roughness length and the displacement height have since remained consistent.". Reference to Cheng et al. (2018) has been added to the manuscript.

  As far as we are aware detailed economic activity, population and traffic data for Beijing is not publicly available. Economic activity surrounding the site is described in section 2.1 as shops and restaurants, data isn't available to add further detail to this, but we feel this is sufficient to enable the reader to interpret the data. As suggested, we have added more detailed descriptions of population density, traffic and the trees present has been added to the manuscript as quoted below.

  "The population density in the Chaoyang district is 7500 people $km^{-1}$ (Amir Siddique et al., 2020)."

  "Jing et al. (2016) classified the major roads present in the footprint as "Artery Roads". The average speed of roads in this classification peaked at ~43 km $h^{-1}$ at 04:00. Day time average speeds peak at ~33 km $h^{-1}$ (12:00) with a minimum average speed of ~28 km $h^{-1}$ (18:00). Average traffic volume on artery roads peaked at ~3100 vehicles $h^{-1}$ (07:00) with a minimum average flow of ~250 vehicles $h^{-1}$ (03:00)."

  "In 2002 dominant tree species in Beijing were *Sophora japonica*, *Populus tomentosa* and *Juniperus chinensis* (Yang et al., 2005). Large scale tree planting occurred between 2012 and 2015 increasing forest cover by 10% with the main species planted being *Pinus tabulaeformis*, *Sophora japonica*, *Fraxinus chinensis*, *Salix spp*. and *Populus spp*. (Yao et al., 2019)."

- **The manuscript reports fluxes collected during two field campaigns, a 28-day campaign in winter and a 40-day campaign in summer. However, after removing the averaging periods affected by lack of turbulence and instruments maintenance, as well as periods not meeting the stationarity criteria, only 25% and 70% of the data, respectively, were used for further analysis. This means that results are based on 7 and 27 net days of flux measurements. For the summer case, an almost one month of flux data might be enough for an initial assessment, but for the winter case, 7 days of data (for some VOC species even less) are not sufficient.**

  **The authors may consider the relevance of including results of the winter campaign in this article. In an urban set up, like Beijing, seven days of flux data may represent nothing else than noise. In addition, many of the analysis performed in this study were limited to the flux data collected during the summer campaign.**

  Response: We agree with the reviewer that the winter data has a high uncertainty, for this reason it was excluded from further analysis.

  As eddy covariance flux measurements of VOCs have not previously been made in Beijing we feel it is worth while to include the winter data here for reference but owing to the uncertainties highlighted we do not draw any firm conclusions from this data set, this has been highlighted in section 4.

- **The impact of regional sources of VOCs in the observed fluxes deserves further analysis and discussion. The authors need to investigate under which synoptic conditions the loss of VOCs (i.e., negative fluxes) was observed. The use of back trajectories may help to this assessment.**

  **Did the APHH project include measurements or modeling of the hydroxyl radical? Information on the levels of OH, NO and $O_3$ would make possible to estimate the residence time of each individual VOC species, and would support the proposed loss of VOCs. Also, it would help to evaluate the capacity of the eddy covariance system to measure fluxes of highly reactive VOCs. As indicated in the original revision, the time taken by an air parcel to reach the top of the tower could be higher than the time needed for the oxidation of such compounds, and thus the measured fluxes would not faithfully represent the emissions from the urban surface.**

  **The OH concentration of $1 \times 10^6$ molecules $cm^{-3}$ used to estimate the oxidation time of the assessed species is too low for a highly polluted atmosphere (see Shirley et al., 2006, for the case of Mexico City, www.atmos-chem-phys.net/6/2753/2006). A higher concentration would reduce the lifetime of the VOCs, and therefore, the eddy covariance flux system mounted at 102 m of height might not be able of measuring fluxes of reactive VOCs.**

  Response:
  In the summer campaign all VOCs showed a net positive flux (line 410 in acpd manuscript) with few averaging period showing deposition. These usually occurred at night time when the magnitude of the flux was low. Deposition was more frequently observed in the winter campaign but as discussed above the high uncertainty in the winter data means that for detailed analysis (for example an analysis of back trajectories) could lead to false conclusions being drawn.

  OH, NO and O3 were measured during the APHH campaigns (Slater et al., 2020; Squires et al., 2020 and Whalley et al., 2020). OH was higher in the summer campaign than in the winter campaign, peaking at $2.8 \times 10^7$ molecule $cm^{-3}$ (Whalley et al., 2020). Using this value, the

monoterpene lifetime would be expected to be ~11 mins. Given the average time for an air parcel to reach the inlet is ~68s this would equate to a ~10% loss of the most reactive species measured. The manuscript has been updated to use the measured OH value.

"The OH concentration in the summer campaign reached $2.8 \times 10^7$ molecules cm$^{-3}$ (Whalley et al., 2020) and the ozone concentration $2.6 \times 10^{12}$ molecules cm$^{-3}$, at these concentrations ~10% of monoterpenes (the most reactive species recorded) would have reacted before the measurement height."

- **A final section providing conclusions and final remarks is missing. Scientific articles are expected to close with a short answer to the hypothesis to test. For example, is the eddy covariance a suitable method to measure urban fluxes of VOCs in Beijing? does the emission inventory is accurate for that particular sector of the city? What is needed to improve the inventory's accuracy?**

   Response: Section 4 provides both a summary of the findings and conclusions and has been retitled to make this clear. The conclusions of this work are presented in this section:
   - Despite high mixing ratios of VOCs emission from the city centre is of comparable magnitude to cities in the UK where VOC mixing ratios are comparatively low. This suggests that the elevated mixing ratios are driven by transport from outside the city.
   - The inventory does not accurately predict VOC emission at local scales.
   - In the summer biogenic emissions make a large contribution to atmospheric chemistry. This will grow more significant as anthropogenic emissions are reduced by policy interventions so modelling studies need to include biogenic emissions from the city centre.

As suggested a short description of what would be needed to improve the accuracy of the inventory has been added:

"Comparison of measured VOC fluxes with those predicted by the emissions inventory showed that the inventory failed to capture VOC emission at this local scale. This failure to predict local scale emission is most likely caused by the scaling of city level activity data to the local scale. In order to accurately predict VOC emissions at a local scale higher resolution activity data will be required."

**Specific comments**

**39-40. Eddy covariance flux towers provide invaluable insight to evaluate the accuracy of gridded emission inventories, but they do not validate emission inventories.**
Response: "validate" removed and changed to "evaluate the accuracy of"

**40-49. The abstract does indicate flux measurements conducted on summer and winter, but results from the latter season are not included.**
Response: As discussed above the winter data set is comparatively small and is not included in the more detailed analysis so was not included here.

**58. '... particulate matter of size smaller than 2.5 μm ...'.**
Response: This has been added to the text

**62-63. This statement needs a reference.**
Response: Reference to Rinsky et al. (1987) added

**78. I would say 'challenging' instead of 'difficult'.**
Response: "difficult" changed to "challenging"

**96. 'Cityscape scale' is a confusing term. Eddy covariance flux towers measure fluxes at neighborhood or district scale.**
Response: "cityscape scale" changed to "district scale"

**97. PTR-TOF has not been defined in the text. Define acronyms the first time they are used in the manuscript.**
Response: "PTR-TOF technology" changed to "proton transfer reaction-time of flight mass spectrometers"

**98. Eddy covariance flux towers measures fluxes, do not estimate fluxes.**
Response: "estimate" changed to "measurements"

**110. See previous comment about the use of flux towers to evaluate the accuracy of emission inventories**
Response: "assess the validity of" changed to "evaluate the accuracy of"

**120. The description of the monitored sector of the city is still incomplete. More information on the economic activities, vehicular traffic volume and composition, population density, number of trees and characteristics, distribution and variability of building heights, etc., is needed to put in context the observed fluxes, and then be able to improve the emission estimates of the gridded emissions inventory.**
Response: See response to previous comment.

**121-123. This sentence is confusing. A 325-m tall tower was used as measuring platform, but the eddy covariance flux system was mounted at a height of 102 m.**
Response: "325 m high" removed

**124. Include the height above sea level of the location where the tower was located.**
Response: the base of the tower is 49 m above sea level, this has been added to the text.

**158. Explain the purpose of the system to evaluate the vertical gradient of VOC concentrations. I may be wrong, but the text does not indicate it had been used for the flux data analysis or discussion.**
Response: The gradient system was designed to monitor VOC gradients at the IAP site. These data do not form part of this study, but a description of the gradient switching system was included as it forms part of the PTR-MS set up.

**289. Do you mean that flux data were flagged if $u* < 0.175$ m s$_{-1}$?**
Response: Yes, this has been made clearer in the text.

"files were also flagged if the mean frictional velocity (u*) was less than 0.175 m s$^{-1}$"

**340-345. Footprints at day and night are quite different. For a thorough assessment of the hourly emissions reported in the gridded emissions inventory is mandatory to evaluate the footprint across the diurnal cycle. Do not ask readers search for such information in another article. The**

**background map provides little information. I cannot identify which sectors correspond to residential estates, and which to commercial and institutional buildings, for instance. Also, I cannot identify which are primary and secondary roads. Can you mark the 3rd and 4th rings surrounding the location?**

Response: The footprints presented in Fig. 4 are campaign averages for the winter and summer campaigns and are intended to give the reader an overview of the flux footprint in both campaigns. We agree that for a proper assessment of the inventory an assessment of the hourly emissions is needed. This was performed and is discussed in section 3.5:

*"First, the footprint for each flux aggregation period and the inventory grid for the corresponding hour of day were aligned by transforming the flux footprint into the coordinate reference system of the inventory."*

We don't feel it adds to the manuscript to include maps of the flux footprint across the diurnal cycle as this would add a lot of figures which convey little new information to the reader.

The background map has been changed to a satellite image to provide more information. Major roads are highlighted in yellow and the 4th ring road is labelled. Commercial properties in this area are mainly shops and restaurants situated amongst residential buildings rather than clear residential or commercial estates. From a map view large enough to display the flux footprint it is difficult to show the use (residential or commercial) of individual buildings but the area is described in section 2.1.

*"The campus is situated north of central Beijing between the 3rd and 4th ring roads, with parkland to the east and west and a mix of dense residential and commercial (restaurants and shops) buildings to the north and south."*

[Figure]

*Figure 4. The campaign mean flux footprint climatologies for the winter (left) and summer (right) field campaigns. The IAP meteorological tower is represented by the red dot and surrounding 100 × 100 m cells are coloured by their mean contribution to the flux. The contour lines correspond to the*

*distances from the tower where the surface contributions to the measured fluxes cumulate to 30, 60 and 90% respectively. Map tile sets are © Google. Map data @ 2020 Imagery @ 2020, CNES/Airbus, Landsat/Copernicus, Maxar Technologies.*

**Table 1. Plots showing the diurnal variability of these VOC species would provide better insight in the context of the study. It does not matter, if a more detailed analysis is presented somewhere else.**

Response: These plots have been added to the manuscript

[Figure]

"Figure 5. Diurnal profiles of the dominant VOCs observed in Beijing during APHH-Beijing winter (blue) and summer (red) campaigns (in ppbv). Error bars show standard deviation across the measurement period."

**422-425. This point deserves further discussion. I would say '… loss of VOCs by chemical reactions in the presence of fresh emitted NO …', instead of '… deposition of VOCs …'. Did the APHH project**

**include measurements/modeling estimates of OH? Knowing the levels of OH, NO, O₃, etc. would make possible to estimate the residence time of each individual VOC species.**
Response: The lifetime of two anthropogenic compounds (benzene and butene) in winter conditions has been assessed and the following text added to the manuscript:

"*In the winter campaign OH concentrations peaked at $8 \times 10^6$ cm$^{-3}$ (Slatter et al., 2020) and the average O3 concentration was 16.4 µg m$^{-3}$ (Shi et al., 2019). Using rate coefficients from Atkinson and Arey (2003) the expected atmospheric lifetimes of benzene and 1-butene with respect to OH would be 1.2 days and 1.1 h, respectively. The lifetimes of benzene and monoterpenes with respect to O3 would be over a year and 6 days, respectively. The lifetime of less reactive compounds such as aromatics is therefore long enough to allow for transportation from outside the flux footprint.*"

**426-428. I do not understand this statement. Do you mean that people cook and clean less during the winter?**
Response: We mean that indoor activities such as cooking and cleaning may impact the external environment less in the winter when people have windows closed than in the summer when ventilation is higher.

"Sources such as cooking and solvents in cleaning products, cosmetics and paints contribute to urban VOC emission (Karl et al., 2018) and these sources are likely to be reduced in the winter when cold weather causes ventilation of buildings to be reduced."

**443. It would be useful to include in the text the variability observed in the fluxes (e.g. ±1 standard deviation) for comparisons to fluxes reported in the literature.**
Response: As suggested this has been added

**Figure 5. I cannot observe any dashed line, and hardly observed the dotted ones.**
Response: The dashed lines (flux without the storage term) and dotted lines (u* filtered fluxes including the storage term and gap filled with the average flux above the u* threshold for that hour) agree very well with the measured flux including the storage term (solid line) so in many places only the solid line is visible. We feel it is helpful to include these lines to show the small impact these the storage term and gap filling data u* flagged data have on the total flux.

**637. More information about how the emission inventory was built is needed. It will help to understand the inventory's overestimations. Does the inventory estimate emissions based on activity data at city scale or use data for each individual grid? Which is the source of the emission factors used to estimate traffic emissions? Were the emission factors locally derived? How were the emission profiles determined? How were the evaporative emission estimated? Are there emission estimates for summer and winter?**
Response: The inventory is based on city scale activity data which is downscaled using proxies such as population and traffic flow data (Zheng et al., 2017). While MEIC is intended as a regional inventory. It's valuable to use it here as it is the most widely used inventory in China, and it has previously been downscaled to the resolutions needed here (Zheng et al., 2017).

The inventory provides monthly emissions data with emissions assumed to be consistent across each month. A diurnal cycle based on local activity data is applied to each emission sector to give a diurnal profile. Emission factors are locally derived with multiple studies reporting VOC emission factors from traffic and evaporative sources in China, these are summarised in a review by Li et al. (2017b) from which the inventory draws data (Zheng et al., 2018).

Further details have been added to the manuscript to make this clearer:

"The emission factors used by the inventory are drawn from a broad range of studies (Li et al., 2014; Zheng et al., 2018) with locally derived emission factors used where possible. These emission factors have been summarised by Li et al. (2017b)."

"Monthly emissions data are provided with emissions assumed to be consistent across each month. A diurnal cycle based on local activity data is applied to each emission sector to give a diurnal profile."

**338. No need to repeat the website hosting the emissions inventory. It was done in the introduction.**
Response: This has been removed

**667-671. Check these two sentences. An overestimation of 3-4% is not equal to overestimate by a factor of 3-4.**
Response: The units here were correct but the values reported by Squires et al. have been updated to give mean values and to provide summer CO values.

"Measured emissions of aromatic compounds were 3% and 4% of those predicted by the inventory for TOL and XYL (low mass and high mass aromatics). Squires et al. (2020) compared $NO_x$ and CO fluxes recorded during the APHH-Beijing campaigns with the MEIC inventory and also observed a similar overestimation of emissions by the inventory, with the inventory overestimating the measured flux 11 and 10 times respectively"

**A marked-up version of the manuscript is presented below with changes highlighted in yellow. In addition to the changes discussed above table 2 was corrected to show stationarity filtered fluxes including the storage term.**

[revised manuscript text omitted]